# Aging-linked systemic lipid signature is reprogrammed by caloric restriction in rhesus monkeys

Salma I Abou Elhassan [iD][1], Josef P Clark[2,3], Di Kuang [iD][2], Timothy W Rhoads [iD][4], Ricki J Colman [iD][5,6], Joshua J Coon[1,7,8,9], Rozalyn M Anderson [iD][2,3][✉] & Katherine A Overmyer [iD][1,7,9][✉]

## Abstract

Caloric restriction (CR) without malnutrition delays aging in diverse species, including primates, with metabolic changes implicated in this process. To facilitate exploration of CR metabolism with aging, we developed a 15-minute LC-MS/MS metabolomics and lipidomics method, leveraging monophasic extractions and wide elution-strength solvents. We analyzed 494 plasma samples collected over 25 years from male and female rhesus monkeys (*Macaca mulatta*) on a Control or CR (30% restricted) diet. Quantitation of 359 biomolecules revealed that aging, followed by sex and diet, had the largest impact on metabolite abundances. In both sexes, aging was associated with significantly lower plasma levels of sphingomyelins (SMs) and higher levels of diglycerides (DGs) and triglycerides (TGs), each of which was opposed by CR. Sex dimorphism was evident by the increased abundance of phosphocholine (PC)-containing lipids in females. These results highlight the utility of a rapid metabolomics and lipidomics approach to elucidate complex biology in large-scale studies.

**Keywords** Metabolomics; Lipid Metabolism; Caloric Restriction; Aging; Rhesus Macaque
**Subject Category** Metabolism

## Introduction

Aging is a multifaceted process marked by systemic changes that increase susceptibility to diseases such as cancer, diabetes, and cardiovascular diseases. Growing evidence indicates that metabolic dysfunction is a critical component of disease vulnerability (Finkel, 2015; Harrington et al, 2023; Nguyen and Corvera, 2024;

Suomalainen and Nunnari, 2024). Many diseases of aging are metabolic in their etiology, including diabetes and cardiovascular disease, while others like, cancer (Gaude and Frezza, 2014; Hsu and Sabatini, 2008) and neurodegeneration (Camandola and Mattson, 2017; Peggion et al, 2024) have a significant metabolic component. In people, whole body metabolism is remarkably stable through adulthood until ~65 years of age, after which a gradual decline in metabolic rate begins and is coincident with the onset of age-related disease vulnerability (Pontzer et al, 2021; Rhoads and Anderson, 2021). Nearly a century ago, the ability of caloric restriction (CR) to delay aging was first documented (McCay et al, 1935), and although the active mechanisms have remained elusive, it is clear that metabolic adaptation is a critical feature in the ability of CR to delay aging and enhance longevity (Anderson and Weindruch, 2010; Balasubramanian et al, 2017a; Longo and Anderson, 2022).

Among nonhuman primates, rhesus macaques (*Macaca mulatta*) are a valuable model for aging due to their high genetic similarity to humans, sharing ~93% of their overall genome sequence and over 97% similarity in protein-coding genes and amino acid sequences (Gibbs et al, 2007). Not only do they share this genetic homology, but they also exhibit lipid and lipoprotein metabolism patterns that are highly comparable to those of humans, along with parallel trajectories of age-related physiological changes (Chiou et al, 2020; Stefanoni et al, 2020). These changes include declines in insulin sensitivity, sarcopenia, and immune system remodeling, making them an especially relevant and translatable model for studying human aging (Shang et al, 2017; Zheng et al, 2014; Pergande et al, 2024; Palliyaguru et al, 2021). Work involving human subjects is inherently complex due to multiple confounders that may directly or indirectly shape measured outcomes, such as lifestyle, diet composition, degree of adiposity, physical activity, alcohol consumption, substance abuse, and ethnicity. In addition to dietary and metabolic interventions, sex dimorphism is increasingly recognized as an important factor in aging biology. Across mammals, males and females frequently exhibit distinct lifespan trajectories, characterized by sex-specific differences in survival, disease susceptibility, and molecular aging

[1]Department of Biomolecular Chemistry, University of Wisconsin-Madison, Madison, WI 53706, USA. [2]Department of Medicine, University of Wisconsin-Madison, Madison, WI 53705, USA. [3]Geriatric Research, Education, and Clinical Center, William S. Middleton Memorial Veterans Hospital, Madison, WI 53705, USA. [4]Department of Nutritional Sciences, University of Wisconsin-Madison, Madison, WI 53706, USA. [5]Wisconsin National Primate Research Center, University of Wisconsin-Madison, Madison, WI 53715, USA. [6]Department of Cell and Regenerative Biology, University of Wisconsin-Madison, Madison, WI 53705, USA. [7]Morgridge Institute for Research, Madison, WI 53715, USA. [8]Department of Chemistry, University of Wisconsin-Madison, Madison, WI 53706, USA. [9]National Center for Quantitative Biology of Complex Systems, Madison, WI 53706, USA. [✉]E-mail: rozalyn.anderson@wisc.edu; kovermyer@wisc.edu

signatures that impact the rate and nature of aging (Hägg and Jylhävä, 2021). These differences have been attributed to hormonal regulation, immune function, and metabolic pathways (Calabrò et al, 2023; Pomatto et al, 2018; Santos-Marcos et al, 2023). The importance of sex as a biological variable has gained traction in the biology of aging studies (Chen et al, 2022) and is expected to provide actionable insights of relevance to human aging.

The complexity of human studies makes rhesus monkeys a good alternative to human subjects, and they are a highly translational model, with genomic, physiological, and behavioral similarities that are shared with humans and a spectrum of age-related diseases and conditions that mirror those prevalent in human aging (Balasubramanian et al, 2017b). CR improves survival and delays the onset of age-related diseases, disorders, and conditions in monkeys (Mattison et al, 2017). Monkeys on CR have lower body weight, lower adiposity, lower fasting glucose and insulin, and greater insulin sensitivity (Colman et al, 2009, 2014); this response is highly similar to that of humans on CR (Das et al, 2017; Kraus et al, 2019). Over the course of the Wisconsin Aging and Calorie Restriction Study, the hazard ratio (HR 1.9) indicated that at any time-point, the control monkeys had almost twice the rate of death when compared to CR animals, and that age-related conditions occurred at more than twice the rate in control animals compared to CR (HR 2.7) (Mattison et al, 2017). Monkeys from this study offer a unique window into the metabolism of aging and the impact of CR on circulating factors linked to aging and disease risk.

To capture metabolic changes with aging, mass spectrometry (MS)-based omics technologies offer an ideal means of profiling accessible biofluids such as plasma or serum. Examples from human clinical studies show a great deal of promise (Fiorito et al, 2024; Liu et al, 2023; Waziry et al, 2023), yet continued analytical efforts are needed to promote MS-based profiling for more routine clinical use. In biomedical research, MS-based omics technologies have been applied for simultaneous monitoring of thousands of biomolecules (Hornburg et al, 2023a; Overmyer et al, 2021; Rensvold et al, 2022). Here, advances in MS technology to increase the depth and range of coverage are truly impressive (Kraus et al, 2025), and several studies report molecular signatures that might serve as disease biomarkers (Pergande et al, 2024; Panyard et al, 2022; Chak et al, 2019). One of the major challenges to analytical methods, however, is that circulating molecules are highly diverse in chemical and physical properties. Accordingly, hydrophilic metabolites (metabolomics) and hydrophobic metabolites (lipidomics) are traditionally analyzed independently and typically involve distinct analytical platforms, e.g., employing hydrophilic interaction liquid chromatography (HILIC) and reversed phase liquid chromatography (LC), sometimes in addition to gas chromatography to cover a broad part of the metabolome (Schwaiger et al, 2019; Rustam and Reid, 2018; Rampler et al, 2018; Baglai et al, 2017). The reliance on separate metabolomics and lipidomics workflows increases sample preparation time, cost, and instrument usage. To date, metabolomics has lagged other omics disciplines due to the challenge in scaling analytical throughput without compromising coverage.

Logistical barriers that have prevented more widespread adoption of MS-derived biomarker panels in clinical diagnostics and tests of treatment efficacy might be overcome with a more rapid yet sensitive method for simultaneous metabolite and lipid quantitation. Towards this end, we developed a unified, high-throughput multi-omics workflow that integrates metabolomics and lipidomics into a single 15-min LC-MS/MS analysis. This method is streamlined from sample preparation to data acquisition

by employing a monophasic extraction (Muehlbauer et al, 2023) and a single chromatographic separation and MS analysis. Unlike traditional workflows, our method eliminates the need for separate analyses, reducing variability, sample processing time, and instrument usage. By minimizing freeze-thaw cycles and standardizing sample preparation, our approach enhances reproducibility and supports large-scale research. We then applied this approach to investigate changes in metabolite and lipid profiles in plasma collected longitudinally from rhesus monkeys undergoing life-long CR as part of the University of Wisconsin–Madison Aging and Caloric Restriction study. We evaluated the impact of diet, sex, and age, and their interactions, on circulating molecular profiles in this cohort. Using our integrated workflow, simultaneous identification of metabolites and lipids revealed both expected and novel signatures of aging, providing new insights into the metabolic effects of CR on longevity.

## Results

### Developing a 15-min LC-MS workflow for integrated small molecule and lipid detection

The first step in creating a comprehensive small molecule and lipid analytical method was to develop a separation approach that would allow for the detection and quantification of a broad range of biomolecules. Given prior success with the Waters HSS T3 column for retaining polar metabolites (Anderson et al, 2024a), we evaluated this chromatography by optimizing: (1) mobile phase composition, (2) separation gradients, (3) resuspension solvents, and (4) loading amounts. To do these evaluations, we assessed the number of detected and quantified biomolecules from a metabolite and lipid extract of NIST1950, a human plasma reference material. Critically, these extractions were done using a monophasic extraction solvent that contains n-butanol and has a high extraction efficiency of biomolecules, from polar metabolites to nonpolar lipids (Muehlbauer et al, 2023).

We began by choosing a mobile phase composition that balances retention across molecular polarities, ensures efficient separation of a wide range of biomolecules, and that successfully separates peptides and lipids on a single chromatographic column (Kraus et al, 2025; He et al, 2021). This MOST (multi-omics single shot) mobile phase system consists of 100% aqueous for mobile phase A and 90% isopropanol/10% acetonitrile for mobile phase B. In contrast to the commonly used mobile phase in the metabolomics field (i.e., 100% methanol for mobile phase B) (Man et al, 2021; Windarsih et al, 2022), the MOST gradient with 90% isopropanol provides higher elution strength for eluting very nonpolar metabolites (lipids). We evaluated with and without ammonium formate in mobile phase B, and we found that ammonium formate was critical for the detection of neutral lipids, such as diglycerides (DGs) and triglycerides (TGs), and had a minimal impact on polar metabolites. Moreover, we tested various gradient profiles, both linear and stepped, to determine the optimum peak shapes and resolutions to improve analyte coverage. We also considered the overall analytical time, with the goal of achieving a rapid 15-minute method. We found that a 6-min linear gradient from 0 to 100% B provided good metabolite coverage and allowed us to achieve our goal (Fig. 1A,B and Appendix Table S1).

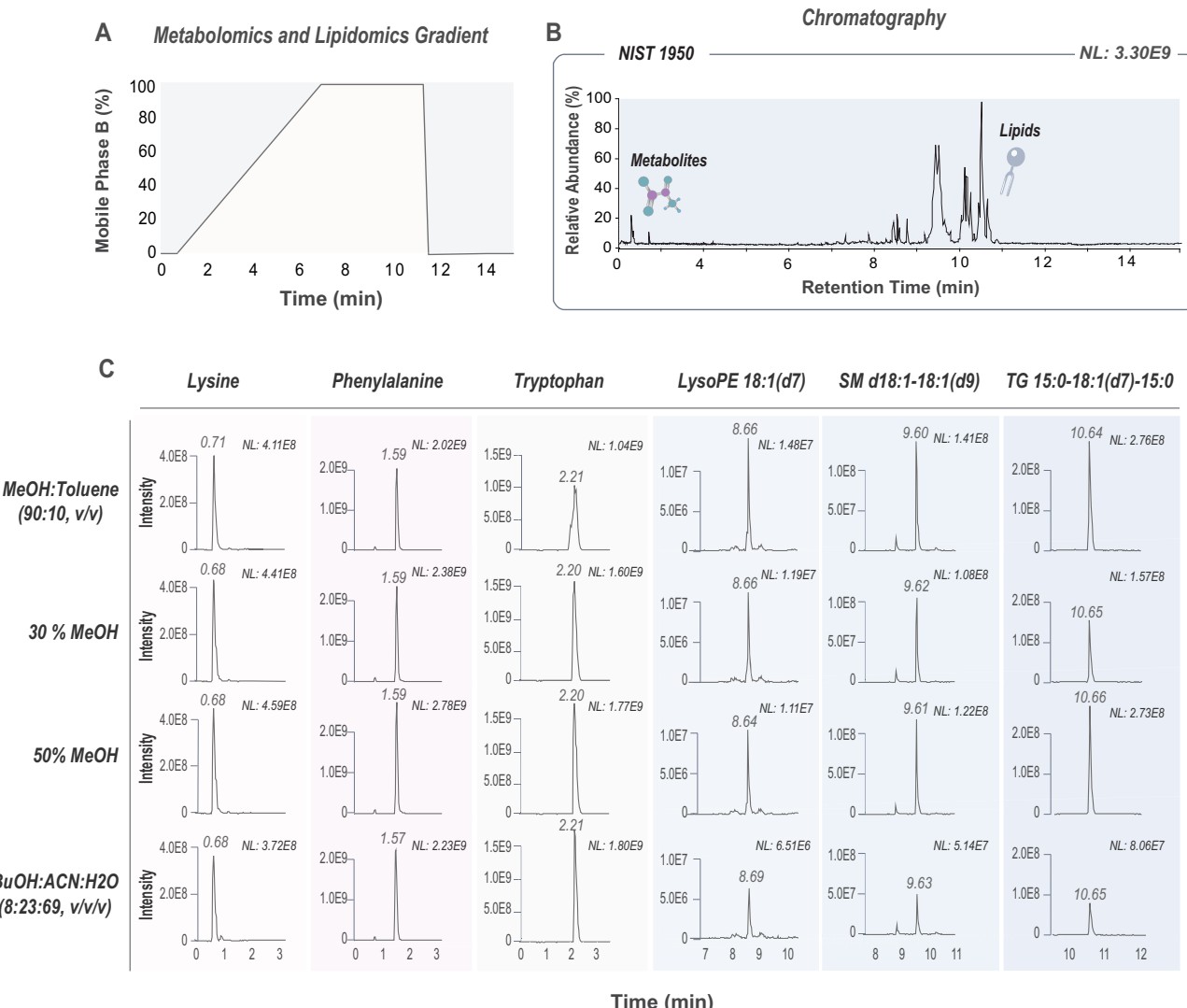

**Figure 1. Optimal parameters for small molecule and lipid analysis.**

(A) A 6-min LC-MS/MS gradient was developed for small molecule and lipid analysis. (B) Representative chromatogram of the method shows metabolite and lipid separation of NIST1950 plasma extracts. (C) Representative ion chromatograms show peak shapes of spiked-in isotopically labeled internal standards (amino acids and Avanti SPLASH lipids) into NIST1950 extracts for evaluation of different resuspension solvents.

Next, to find the optimal resuspension solvent for maintaining peak shapes across the gradient, we tested four different resuspension solvents: (a) 9:1 methanol:toluene, (b) 30% methanol, (c) 50% methanol, and (d) 8:23:69 n-butanol:acetonitrile:water. These solvents have been used in previous metabolomics and lipidomics applications for their ability to strike a balance in solubilizing molecules with varying hydrophobicities (Danne-Rasche et al, 2018; Overmyer et al, 2021). To assess their performance by evaluating peak shapes, we spiked in isotopically labeled internal standards, amino acids and Avanti SPLASH lipids, into NIST1950 extracts that were resuspended in the different solvents. Representative ion chromatograms for individual standards are shown in Fig. 1C. Among the tested solvents, methanol:toluene (90:10, v/v) was optimal for lipid recovery, yielding the highest signal intensities for lipids. However, it resulted in lower abundance and low-quality

peak shapes of small polar metabolites. In contrast, n-butanol:acetonitrile:water (8:23:69, v/v/v) provided better performance for small polar metabolites, maintaining good peak shapes and reasonable intensities. However, it was the least effective for lipids, leading to the weakest signals across all lipid standards, likely due to the high water content of this resuspension solvent. Overall, we found that 30% methanol and 50% methanol served as balanced solvents, providing moderate performance for both metabolite and lipid standards. However, 50% provided the highest overall intensities across all metabolite and lipid standards, making it the most suitable choice for achieving robust and balanced recovery in simultaneous metabolite and lipid analysis.

Finally, to evaluate sensitivity and detection limits, we examined a dilution of internal standards, $C^{13}$ $N^{15}$ amino acids and Avanti SPLASH mixture, in a background of NIST1950 extracts; each

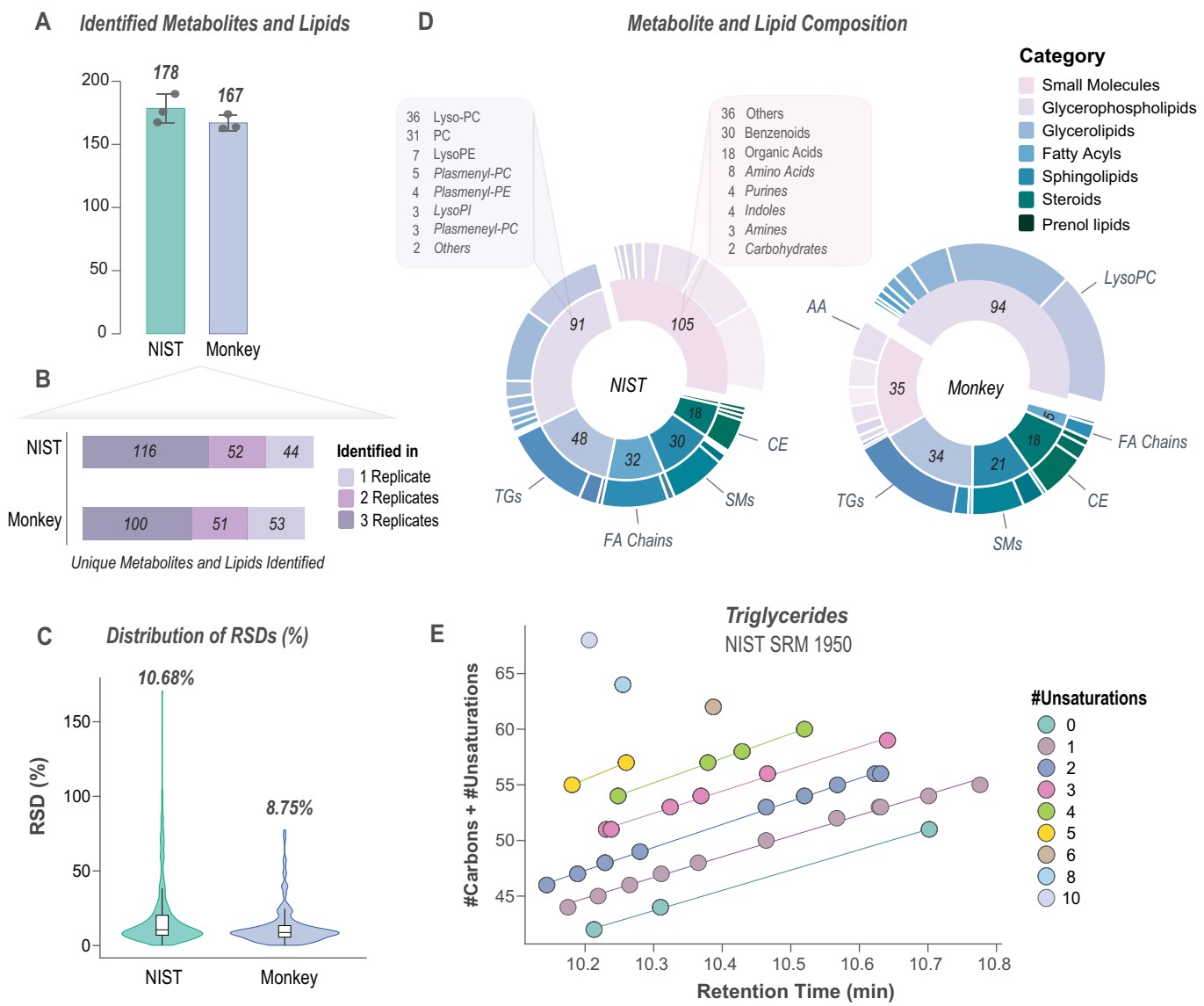

**Figure 2. Evaluation of the combined metabolomics and lipidomics method on human plasma and monkey serum samples.**

(A) The average number of metabolites and lipids identified in NIST1950 and monkey serum samples is 178 and 167, respectively. (B) Barplot shows the number of unique features identified across three technical replicates of NIST1950 and monkey serum samples. (C) Relative standard deviations (RSDs) of identified metabolites and lipids across replicates for each sample type are 10.68 and 8.75% for NIST1950 and monkey serum, respectively. (D) We show the category and main class distribution of identified metabolites and lipids in NIST1950 and monkey serum samples. (E) Scatter plot of triglycerides' effective carbon number relative to retention time found in NIST1950 plasma replicate samples shows expected linearity. Data information: error bars denote standard deviation, and dots represent the number of technical replicates ($n = 3$). For boxplots, the middle horizontal line represents the median, and box margins are first and third quartiles, with vertical lines extending ±1.5-times the interquartile range. Source data are available online for this figure.

internal standard varied in concentration within the stock mixture (see Methods and Appendix Table S2). We found that most amino acids were detected with linear response from 50 nM to 45 mM, representative amino acids are plotted in Appendix Fig. S1A. For lipids, we detected 8 out of 12 lipids at the highest concentration tested. Among these detected lipids, phosphatidylcholine (PC) 15:0/18:1(d7), sphingomyelin (SM) 18:1/18:1(d9), and triglyceride (TG) 15:0/15:0/18:1(d7) maintained linear response over a range of 0.4 to 200 µM (Appendix Fig. S1B). In the extract of NIST1950, we found that, on average, amino acids and lipids fell within the linear response range of the internal standards.

To assess the robustness and reproducibility of our 15-min LC-MS/MS method, we evaluated quantitative performance with extracts of NIST1950 human plasma, a well-studied, qualified reference standard for metabolomics and lipidomics research. In addition, given our goal of eventually analyzing a large cohort of monkey (rhesus macaque) plasma samples, we compared the method's analyte coverage between technical replicates of NIST1950 and a pooled monkey serum sample, purchased from BioIVT (Fig. 2A).

We first assessed the number of features identified in triplicate of NIST1950 and pooled monkey serum samples using Compound

Discoverer 3.3 (Thermo Fisher Scientific) and LipiDex 2 (Anderson et al, 2024b). These annotations were not subjected to extensive manual validation for isomer or putative biomolecules. For this work, we only considered Level 1 and Level 2 annotations (Schrimpe-Rutledge et al, 2016); Level 2 annotations were enabled by data-dependent acquisition (DDA) of MS2 spectra. We identified an average of $178 \pm 12$ features in NIST1950 and $167 \pm 6$ in pooled monkey serum samples (Fig. 2A). Among these features, there was a high degree of overlap across replicates, with 116 and 100 unique features consistently detected in all NIST1950 and monkey serum replicates, respectively (Fig. 2B). Moreover, the reproducibility of our method was verified by low relative standard deviations (RSDs) across replicates, with median RSDs of 10.68 and 8.75% for human plasma and monkey serum, respectively (Fig. 2C). Correlation of biomolecule abundance between replicates was high (average $R^2 = 0.94$) (Appendix Fig. S2A). These findings emphasize the reliability of our method, even when applied to different biological matrices.

To understand the compound class distribution of these annotations, we assigned categories and main classes based on the HMDB chemical taxonomy (Wishart et al, 2022) and the LIPID MAPS classification system (Fahy et al, 2005) (Fig. 2D). Of the compounds detected in NIST1950, 32.2% were small molecules comprised of benzenoids, organic acids, and amino acids, among other classes. The second highest category was glycerophospholipids (27.9%), comprised predominantly of choline-containing lipids. Glycerophospholipids were highly detected in monkey serum (44.8%), and this was the largest main class detected in this sample, followed by small molecules (16.9%). Surprisingly, we detected roughly twice as many small molecules in NIST1950 human plasma compared to monkey serum. Notably, many of the small molecules identified only in NIST1950 were consistent with exposome-related compounds, such as drug metabolites, plasticizers, pesticides, and personal care product components (i.e., sunscreen and cosmetics ingredients), and these molecules are not expected in the monkey serum. We summarized the HMDB-listed "sources" and "roles" of these small molecules in Dataset EV1. We also detected a large number of TGs in both human NIST1950 plasma and monkey serum ($n = 38$ in NIST1950; $n = 29$ in monkey serum). Figure 2E highlights the effective carbon number (carbon number + number of unsaturations) relative to retention time for the TGs found in NIST1950. These data showcase an expected linear trend, providing further confidence in these TG annotations.

## Longitudinal plasma metabolomic and lipidomic profiling of rhesus monkey reveals age as the primary determinant of molecular variation

To further evaluate the robustness of our multi-omics method and its use for large-scale studies, we applied it to a large cohort of rhesus monkey plasma samples collected over 30 years from the University of Wisconsin–Madison Aging and Caloric Restriction study. This cohort included 76 monkeys, evenly divided between control-fed (Control, $n = 38$) and caloric-restricted (CR, $n = 38$) groups. Monkeys were enrolled in the study at adulthood and monitored through their entire lifespan. Plasma samples were collected biannually from each individual, and for this analysis, specimens taken at 2–5 year intervals across the duration of the

study were selected as indicated in Fig. 3A. A total of 494 specimens were analyzed in this work, which is the first exploration of the long-term longitudinal changes in the metabolome and lipidome of these monkeys.

To monitor the method's performance across this dataset, we used quality control (QC) pooled monkey serum samples (same as above, see Fig. 2) run among study samples during data collection. In total, we ran ~650 samples, including study samples and QC samples, over 10 days (90 samples per day) in 12 batches. The data were processed together in Compound Discoverer 3.3 (Thermo) and LipiDex 2 and normalized for batch effects based on median feature intensities. Normalized feature abundances in the QC samples showed minimal variability (Average intra-batch RSD = 17%, overall RSD = 31%), underscoring the method's reproducibility for large-scale studies (Fig. 3B; Appendix Fig. S3). Across the dataset, we annotated 196 metabolites and lipids (Level 1 and Level 2 identifications). These annotated features showed a clear linear relationship between retention time (RT) and mass-to-charge ratio ($m/z$), with compounds from the same category clustering together (Fig. 3C). Small molecules primarily eluted in the first 6 min, while lipids primarily eluted in the last 4 min of the gradient. These data demonstrate that our method was robust and suitable for the quantification of metabolites and lipids across hundreds of samples.

Next, we examined the biological variation captured by the method. To do this, we performed principal component analysis (PCA). Principal component 1, capturing 28.8% of the variance in the data, revealed a distinct age-related separation with samples from early life stages clustering separately from those of late life stages (Fig. 3D). This separation underlines the strong influence of age as the primary covariate driving metabolic and lipidomic variation in this dataset. Evaluating other covariates of sex and diet, we found modest associations of sex with principal component 2 (12.2% of the variance) and between diet and principal components 1 and 2 (Fig. EV1A,B). When we plotted principal component 1 over time and evaluated the effect of caloric restriction, we found a subtle shift in the regression model based on diet (y-intercept$_{|CR-C|}$ = 0.87, slope$_{|CR-C|}$ = 0.03) but not sex (Fig. EV1C,D). These analyses suggest that delayed aging was reflected in the metabolic profile in CR monkeys compared to Controls, but the difference was obscured by the dominant effect of age.

## Caloric restriction modulates neutral lipids' abundance with aging

Taking an alternate approach to investigate how CR influences lipid and metabolite levels, we conducted mixed-effect multiple linear regression analyses. These models assessed the relationship between each quantified feature intensity and three key factors: Diet (CR vs. Control), sex, and age. With this approach, we were able to evaluate the effect of CR in the context of aging, or said another way, at any given age, we are evaluating the effect of CR. To account for biological variability between individual monkeys, we included monkey identifications as random effects, ensuring that inherent differences across subjects are accounted for in the analysis. Given the inclusion of random effects in these models, likelihood ratio tests were used to calculate $p$ values and effect sizes. All $p$ values were corrected for multiple hypothesis testing using the Benjamini–Hochberg approach.

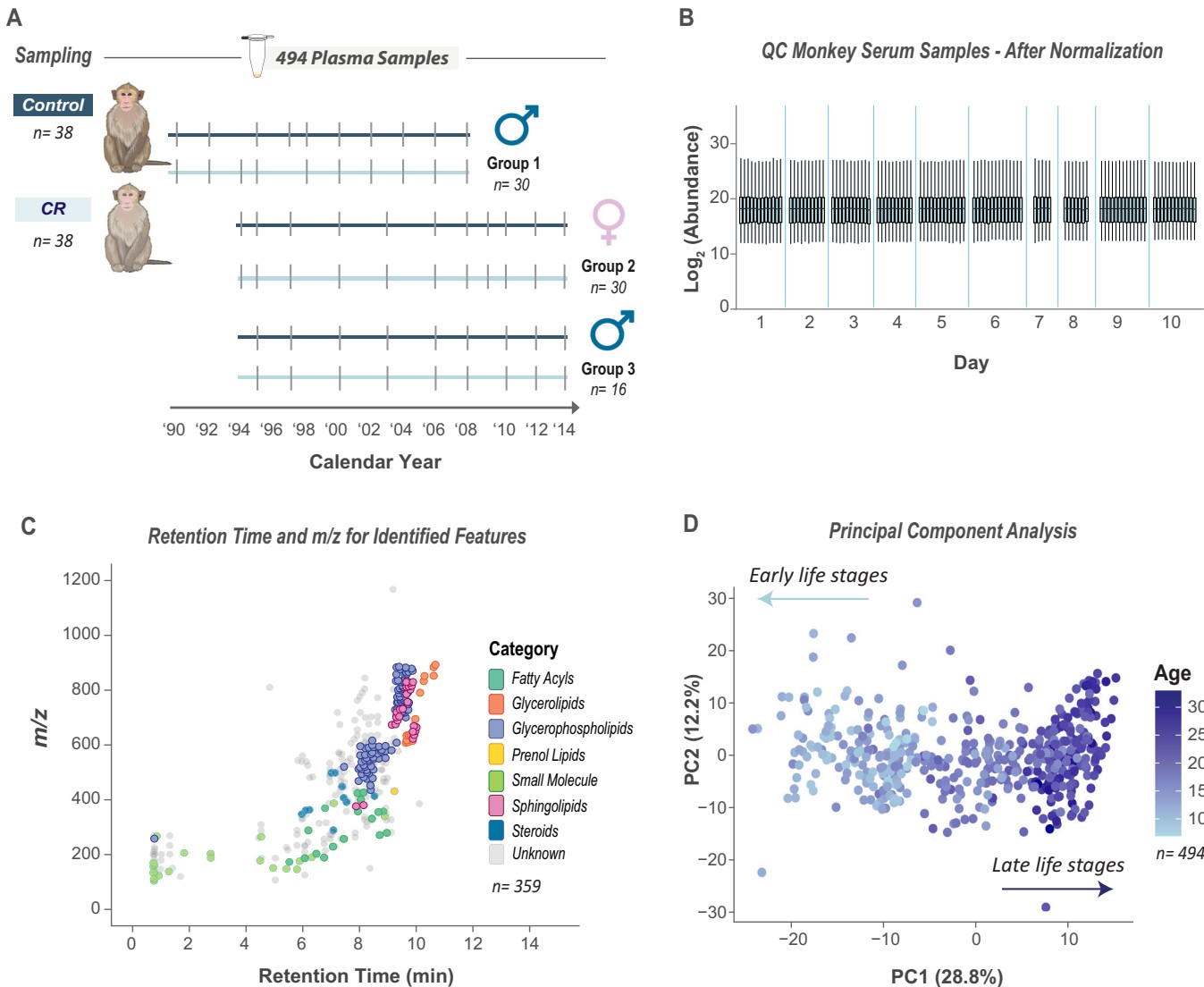

**Figure 3. Measuring metabolites and lipids in longitudinally collected plasma samples from monkeys.**

(A) Longitudinal study design of 76 monkeys, including an initial group of 30 males in 1989, followed by an additional 30 females and 16 males added in 1994. Sampling points (hash marks) are shown for the caloric-restricted (CR) ($n = 38$) and Control-fed ($n = 38$) monkey groups. (B) Instrument performance was assessed using quality control (QC) pooled monkey serum samples during 10 days of data collection. (C) Scatter plot shows elution behavior of identified (in color) and unidentified (in gray) metabolites and lipid ($n = 359$). (D) PCA of study samples using quantitative values from omics data (lipids and small molecules, $\log_2$ transformed and centered around 0 for $n = 494$ monkey plasma samples), shows that principal components 1 and 2 capture 29 and 12%, respectively, of the variance between monkey plasma samples; dots colored based on the age of monkeys when samples were collected. Data information: For each boxplot, the middle horizontal line is the median, and box margins are first and third quartiles, with vertical lines extending ±1.5-times the interquartile range. Source data are available online for this figure.

Our analysis revealed significant differences in lipids between CR and Control monkeys, particularly SMs, DGs, and TGs (Fig. 4A). SMs were significantly more abundant in circulation in CR monkeys, whereas DGs and TGs were significantly lower in abundance, indicating that discrete aspects of lipid metabolism are responsive to the CR diet. We next performed rank-based enrichment analysis using categories, such as compound class, fatty acyl unsaturation level, and odd *vs*. even-chain fatty acyl composition. As might be expected from the volcano plot, SMs were the most significantly enriched lipid class in CR (Fig. 4B),

while TGs, DGs, and ceramides (Cer[NS]) were negatively enriched in CR relative to Controls. The effect of diet for specific classes is further visualized in Fig. EV2A. Interestingly, previous studies investigating circulating factors linked to insulin resistance reported a predictive model for loss of insulin sensitivity based on DG composition that was independent of adiposity and effective in advance of hyperglycemia (Polewski et al, 2015). Furthermore, in previously published data on the monkey cohort presented in this study, fasting glucose levels in control monkeys begin to elevate around 23 years of age, indicating shifts towards insulin resistance

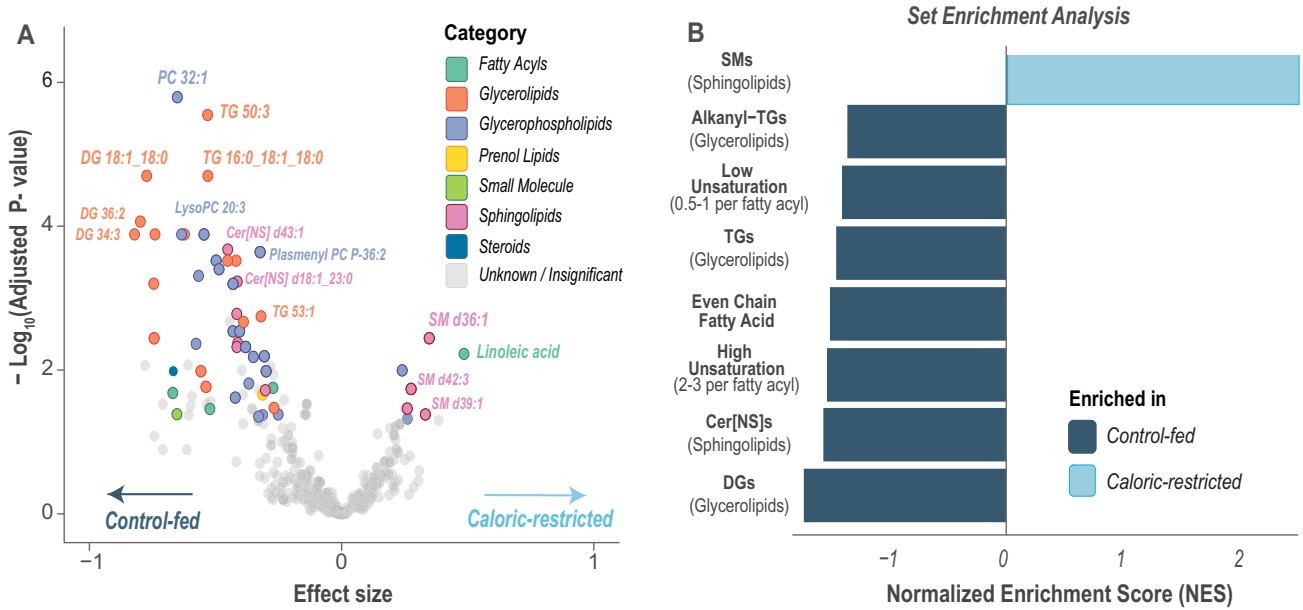

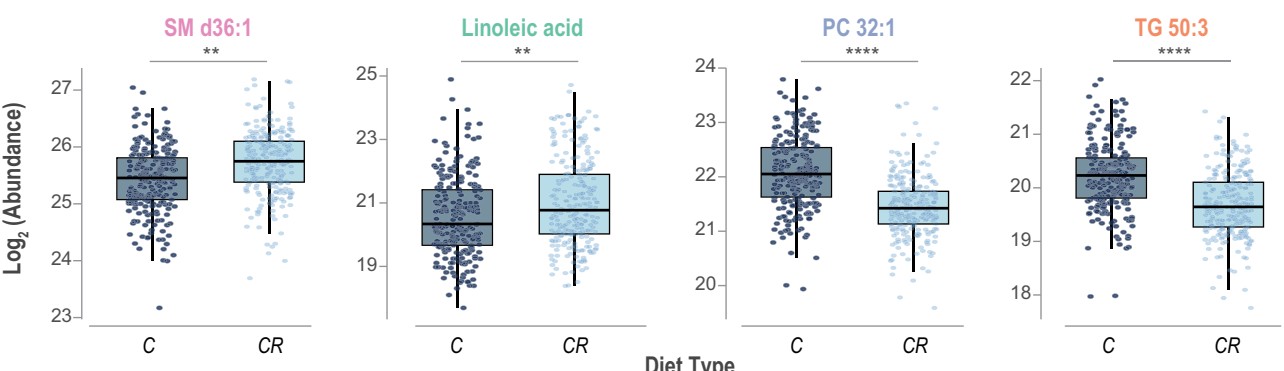

**C**  *Abundance of Top Significant Biomolecules in Caloric-restricted (CR) vs. Control-fed (C)*

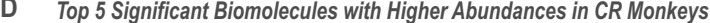

**D**  *Top 5 Significant Biomolecules with Higher Abundances in CR Monkeys*

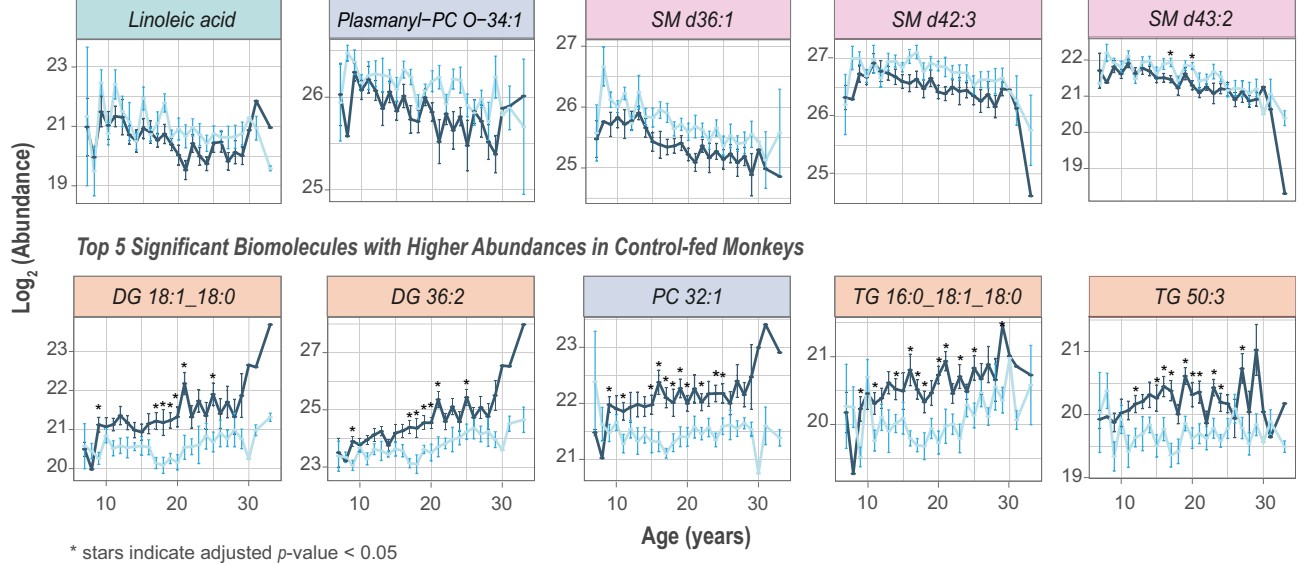

*Top 5 Significant Biomolecules with Higher Abundances in Control-fed Monkeys*

* stars indicate adjusted *p*-value < 0.05

◄

**Figure 4. Lipidomic and metabolomic differences between caloric-restricted (CR) and control-fed (C) monkeys.**

(A) Associations of biomolecules ($n = 359$) with diet type (CR vs. Control) were determined using linear regression models. Statistical significance was evaluated using likelihood ratio tests ($\chi^2$ ANOVA), and $p$ values were adjusted for multiple testing using the Benjamini–Hochberg method. The $-\log_{10}$ adjusted $p$ values are plotted against the effect size; biomolecules with an adjusted $p$ value <0.05 ($n = 58$) are colored based on their category, with non-significant and unidentified features ($n = 301$) shown in gray. (B) Rank-based enrichment plot based on the linear regression analysis shows the top enrichment categories with normalized enrichment scores (NES). (C) Comparison of biomolecule abundance among all CR and C samples ($n = 494$, 258 caloric-restricted and 236 control-fed samples) highlights the top four significantly differentially abundant biomolecules (adjusted $p$ value <0.05). To represent both directions of association, the two most positively associated (e.g., SM d36:1 and linoleic acid) and two most negatively associated (e.g., PC 32:1 and TG 50:3) biomolecules with diet are labeled, with adjusted $p$ values of $3.61 \times 10^{-3}$, $5.96 \times 10^{-3}$, $4.47 \times 10^{-9}$, and $1.58 \times 10^{-8}$, respectively. (D) Biomolecule abundances of lipids significantly associated with diet are plotted as a function of age; title colors are based on the biomolecules' category. Data information: For (A), statistical significance for the volcano plots was assessed using log likelihood ratio tests ($\chi^2$), and $p$ values were adjusted for multiple testing hypothesis using the Benjamini–Hochberg method. For (C) ** and **** indicate $p < 0.01$ and $p < 0.0001$, respectively, as calculated with ANOVA. For each boxplot, the middle horizontal line is the median, box margins are first and third quartiles, with vertical lines extending ±1.5-times the interquartile range, and each dot represents an individual sample. For (D) dots show the mean abundance, error bars show the standard error of the mean, and stars indicate adjusted $p$ value <0.05 using Welch's unpaired $t$-test. Source data are available online for this figure.

(Mattison et al, 2017). Glucoregulatory impairment was also monitored by regular glucose tolerance tests in these monkeys, and control, but not CR, showed glucoregulatory impairment as early as 7 years of age. Notably, elevations in DGs and TGs appear earlier than elevations in fasting glucose (~20 years), and track closely with elevations in adiposity in the control vs. CR. Elevations in DGs and TGs frequently co-occur with insulin resistance (Ormazabal et al, 2018), and our data showing elevated DGs and TGs in control monkeys is in concordance with this observation. In addition to TGs and DGs, a role for SM in the transition from healthy to metabolic syndrome in monkeys was identified by Smith et al, (2022), indicating that lipid signatures could have utility in indexing metabolic health status.

The enrichment analysis also indicated a significant negative enrichment of low-unsaturated (0.5–1 per sn position) and high-unsaturated (2–3 per sn position) lipids in plasma from CR monkeys in our cohort. This enrichment of high fatty acyl unsaturation with CR is especially observed in lipid classes that showed no overall directionality in abundance towards either diet group, i.e., lysoPC and plasmenyl-PCs (P-PCs) (Fig. EV2B). Importantly, diet composition was identical for Control and CR animals, indicating that differences detected here are due to differences in innate lipid handling as a result of lower caloric intake. We further assessed carbon chain evenness and oddness and found that even-chain lipids were significantly less prevalent in CR monkeys compared to Controls. These data identify a CR-induced shift in lipid metabolism, suggesting that lipid metabolic status may contribute to CR's effect on health and survival.

Looking at individual top-ranked species in the response to CR, we found SM d36:1 and linoleic acid (18:2n-6) had higher abundances in CR monkeys compared to Controls (Fig. 4C). Abundance of circulating linoleic acid has been positively associated with insulin sensitivity in people (Beyene et al, 2021; Vessby et al, 1994), and previous studies in mice show its consistent increased circulating abundance with CR throughout adult lifespan, both as a free fatty acid and as a component within other lipid classes (Miller et al, 2017). In contrast, PC 32:1 and TG 50:3 were lower in CR compared to Control. Higher levels of PC 32:1 have been previously reported to be positively associated with type II diabetes (Floegel et al, 2013), a condition that CR protects against in this cohort. To understand the relationship between these CR-significant biomolecules and age, we plotted their abundance at different age points (Fig. 4D; Appendix Fig. S4). Some lipid species

belonging to PCs and lysoPCs, showed minimal age-related changes but a consistent difference between CR and Controls. Other lipids like Alkanyl-TGs, DGs, and TGs were detected as increasing in abundance as a function of age; however, the trajectory of change was more gradual for CR compared to Controls (Fig. EV2C). Some lipids, mainly SMs, such as SM d36:1, SM d42:3, and SM d43:2, decreased with advancing age but with sustained higher abundance in the CR monkeys compared to Controls (Figs. 4D and EV2C). Yet other lipids like linoleic acid and plasmanyl-PC O-36:1 were refractory to age but were consistently more abundant in CR monkeys vs. Controls (Fig. 4D).

## Sex dimorphism is observed in the abundance of phosphatidylcholine lipids

To investigate sex-based metabolic differences in lipid and metabolite profiles, we used a linear regression approach. Sex effects, while also considering dietary effects and age, were evaluated using mixed-effect multiple linear regression analysis. Here, we found 50 annotated biomolecules that were significantly greater in abundance in females compared to males, and six annotated biomolecules that were significantly more abundant in males vs. females. (Fig. 5A). Rank-based enrichment analysis showed categories that were significantly enriched in females included very high and high-unsaturated lipids (>3 and 2–3 unsaturations per fatty acyl, respectively), and choline-containing phospholipids, e.g., PCs and plasmenyl-PCs (Figs. 5B and EV3A). In general, females had a higher abundance of glycerophospholipids, suggesting potential differences in lipid transport between sexes. A selection of significantly differentially abundant biomolecules is presented in Fig. 5C, including testosterone and tryptophan that were higher in males, and plasmenyl-PC P-40:7 and PC 40:7 that were higher in females.

Looking at these sex-specific biomolecules and lipid classes over time, we find that most are changing with age (Figs. 5D and EV3B; Appendix Fig. S5). Many are decreasing over time, including tryptophan, testosterone, and lysoPC 16:0. For several species for which abundance was sex dimorphic, including lysoPC 16:1, PC 38:6, PC 40:7, and plasmenyl-PC P-40:7, an early-adult-life decrease in abundance was detected in males only. This male-specific decrease could be due to a prolonged response to the introduction of the study diet that was matched in composition for

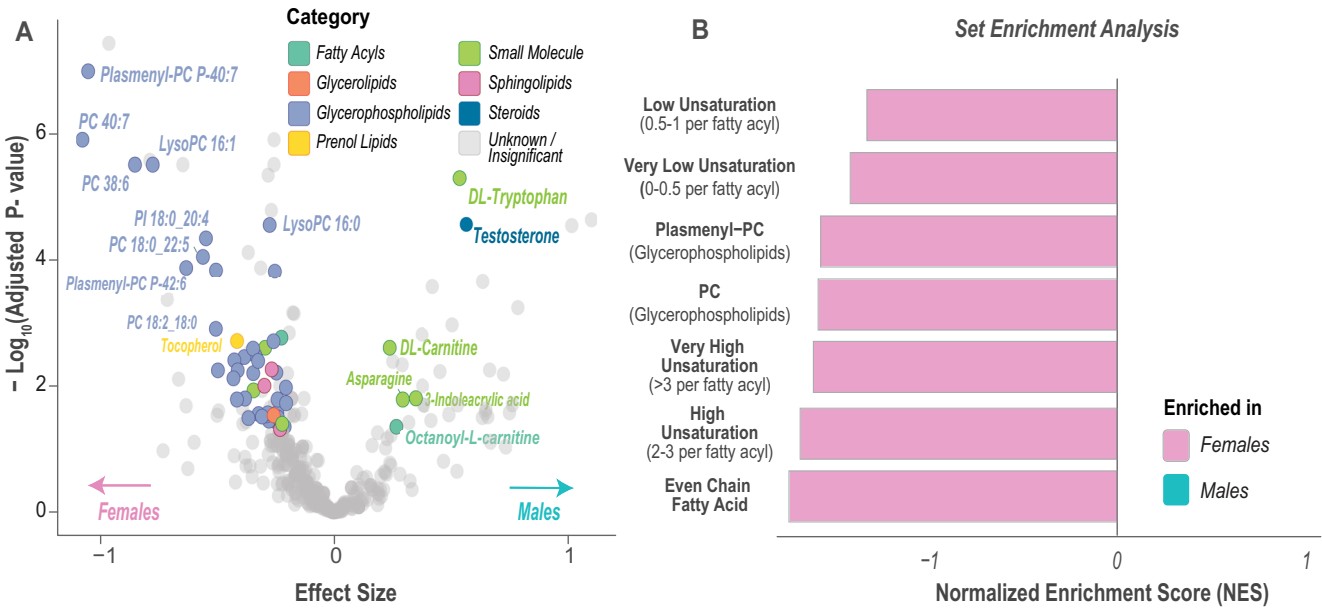

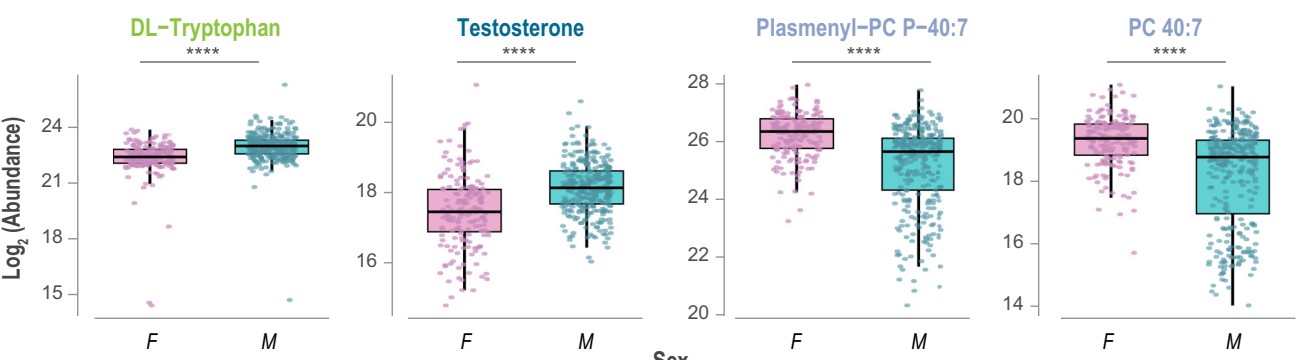

**C** *Abundance of Top Significant Biomolecules in Males (M) vs. Females (F)*

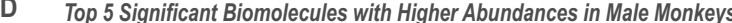

**D** *Top 5 Significant Biomolecules with Higher Abundances in Male Monkeys*

*Top 5 Significant Biomolecules with Higher Abundances in Female Monkeys*

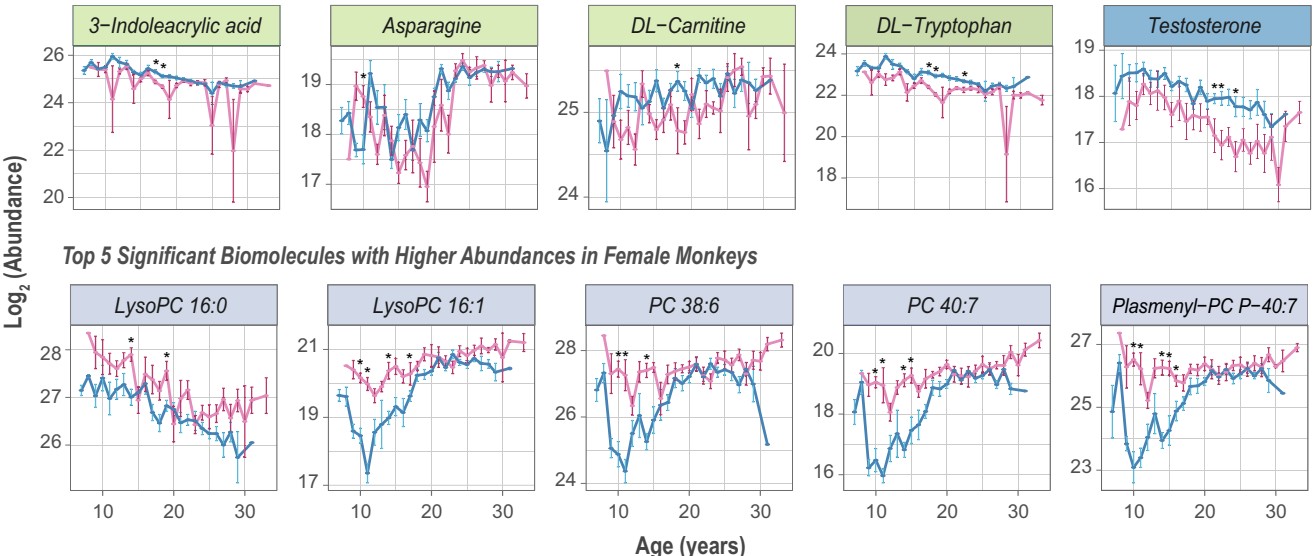

*\* stars indicate adjusted p-value < 0.05*

Figure 5. Lipidomic and metabolomic differences between male (M) and female (F) monkeys.

(A) Associations of biomolecules ($n = 359$) with sex (M vs. F) were determined using linear regression models. Statistical significance was evaluated using likelihood ratio tests ($\chi^2$ ANOVA), and $p$ values were adjusted for multiple testing using the Benjamini–Hochberg method. The $-\log_{10}$ adjusted $p$ values are plotted against the effect size; significant biomolecules ($n = 57$) with an adjusted $p$ value <0.05 are colored based on their category, with non-significant and unidentified features ($n = 302$) shown in gray. (B) Rank-based enrichment was based on the linear regression results, and the top enrichment categories are plotted as normalized enrichment scores (NES). (C) Comparison of biomolecule abundance among all M and F samples ($n = 494$, 316 males and 178 female samples) highlights the top four significant biomolecules (adjusted $p$ value <0.05). To represent both directions of association, the two most positively associated (e.g., DL-tryptophan and testosterone) and two most negatively associated (e.g., plasmenyl-PC P-40:7 and PC 40:7) biomolecules with sex are labeled, with adjusted $p$ values of $1.0 \times 10^{-7}$, $1.22 \times 10^{-6}$, $1.53 \times 10^{-7}$, and $1.12 \times 10^{-6}$, respectively. (D) Biomolecule abundances of lipid species significantly associated with sex are plotted as a function of age; title colors are based on the biomolecules' category, and dots show the mean abundance. Data information: For (A), statistical significance for the volcano plots was assessed using log likelihood ratio tests ($\chi^2$), and $p$ values were adjusted for multiple testing hypothesis using the Benjamini–Hochberg method. For (C), **** indicates $p < 0.0001$ as calculated with ANOVA. For each boxplot, the middle horizontal line is the median, box margins are first and third quartiles, with vertical lines extending ±1.5-times the interquartile range. For (D), error bars show the standard error of the mean, and stars indicate adjusted $p$ value <0.05 using Welch's unpaired $t$-test. Source data are available online for this figure.

Controls and CR but distinct from regular chow. It is not clear why it was detected in males only, and the basis for the shared pattern of response among quite distinct lipid species is not known. Alternatively, the higher abundance of PCs and plasmenyl-PCs in females in early life could be linked to reproductive hormones, such as estrogen (Fig. EV3B). Estrogen is known to regulate phospholipid metabolism, specifically by activating the phosphatidylethanolamine $N$-methyltransferase (PEMT) enzyme essential for PC synthesis (Resseguie et al, 2007). However, it is challenging to put these data into the context of menopause in this cohort, given the current data where direct links to hormone status were not explored. Overall, the observed sex dimorphism in lipid and metabolite profiles, including the higher abundance of testosterone in males and specific choline-containing phospholipids (e.g., PCs and plasmenyl-PCs) in females, strongly suggests a role for biological sex in driving these circulating metabolic signatures.

## Age-associated changes of lipidomic and metabolomic profiles are modulated by diet and sex

Due to the strong influence of age on the biomolecule abundance, we assessed both the impact of age and the interactions between age, diet, and sex using the linear regression approach. We found 312 biomolecules significant with age (Fig. 6A). The biomolecules most increasing with age include LysoPC 20:5, azelic acid, and PI 18:0_20:4. These biomolecules elevated with age were enriched in phosphatidylcholines and highly unsaturated lipids (Fig. 6B; Appendix Fig. S6A). Biomolecules that were decreasing with age include lysoPI 18:0, aspartic acid, and suberic acid. Overall, these biomolecules that were reduced with age were enriched in sphingomyelins, lysoPC, and very low-unsaturated lipids. Notably, this decline in LysoPCs with aging has been reported in previous studies on human subjects. Pan et al, (2023) reported that lysoPCs decline in the plasma of older Chinese adults, indicating their potential as aging biomarkers. Similarly, in the Baltimore Longitudinal Study of Aging, lower plasma lysoPC levels were associated with impaired mitochondrial oxidative capacity in skeletal muscles (Semba et al, 2019).

As shown in Fig. 4D, many significant molecules with diet also had changes with age. Figure 6C highlights 20 biomolecules with a significant age-diet interaction; these include DGs, which have a significantly higher change with aging in control-fed monkeys relative to the CR monkeys. These are in contrast to lysoPCs (19:0, 20:0, and 24:0), which are increasing more in CR monkeys with age

compared to control-fed monkeys. This interaction term can be interpreted as differences in slope between control-fed and CR monkeys with respect to age. The biomolecules that had higher slopes in CR monkeys with age were enriched in sphingomyelins, lysoPCs, and low and very low-unsaturated lipids (Fig. 6D; Appendix Fig. S6B). While molecules with reduced slopes in CR monkeys with age were enriched in DGs and Alkanyl-TGs.

We also evaluated how sex influenced biomolecules' aging profiles. To do this, we used linear regression models and evaluated the significance of the interaction term between age and sex. Biomolecules' age trajectories that are significantly influenced by sex are highlighted in Fig. 6E. In this volcano plot, biomolecules with positive effect sizes indicate molecules that have a higher rate of change with age in males compared to females. An example of this is plasmenyl-PC P-40:7 that has a higher change with age in males, despite its lower abundance in males than in females (see also Fig. 5D). Other biomolecules that have elevated change in males include PC 40:7, plasmenyl-PC P-42:6, and PC 38:6. These biomolecules with positive age–sex effect sizes were enriched in plasmenyl-PCs, PCs, and high-unsaturated lipids (Fig. 6F). In contrast, biomolecules that showed higher rates of change in females with age include linoleic acid, choline, aspartic acid, and acyl carnitines (ACs) 18:1, 18:2, and 16:0; these biomolecules were also enriched in fatty acyls, lysoPCs, and very low-unsaturated lipids (Fig. 6E,F; Appendix Fig. S6C).

## CR modifies biomolecular correlation networks

To explore the association among biomolecules as a function of diet, we performed a Kendall's Tau ($\tau$) correlation analysis on biomolecules in control-fed monkeys ($n = 236$ samples) and in CR monkeys ($n = 258$ samples). Correlation matrices were visualized as heatmaps for Control and CR monkeys separately, with hierarchical clustering of the Control matrix imposed on the CR matrix to enable comparison across correlations (Fig. 7A). There was an overall similarity in the pattern of correlations between Control and CR samples, and biomolecules like SMs (cluster 12), lysophospholipids (clusters 3 and 11), and TGs and DGs (cluster 6) exhibited high within-class positive correlations. Notably, the lysophospholipid clusters, enriched in lysoPIs and lysoPCs, were strongly anti-correlated with clusters containing plasmenyl-PCs, PIs, DGs, and PCs (clusters 7 and 10). This anti-correlation is likely driven by age-induced changes. PCs were among the most significant biomolecules increasing with age, while lysoPCs were

## Age

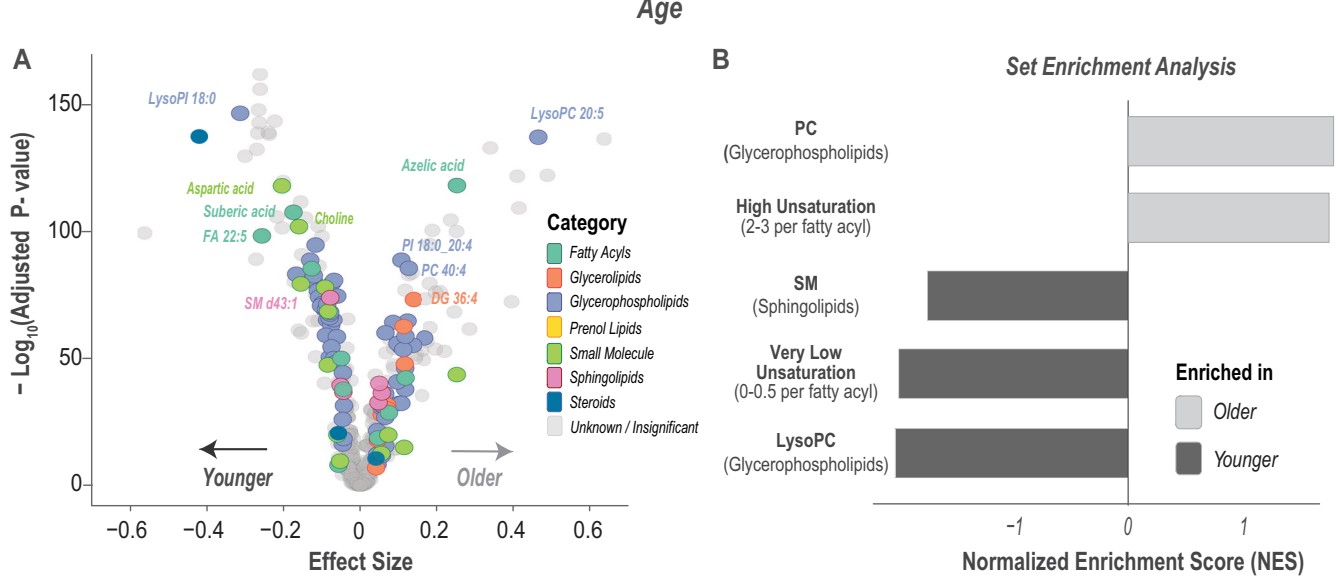

## Age-Diet Interaction

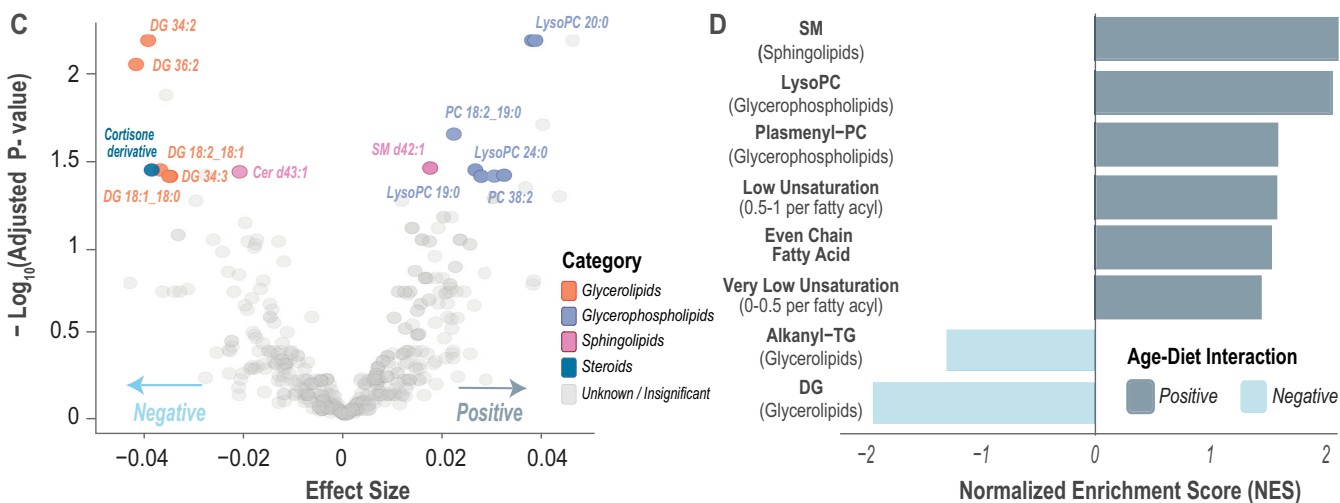

## Age-Sex Interaction

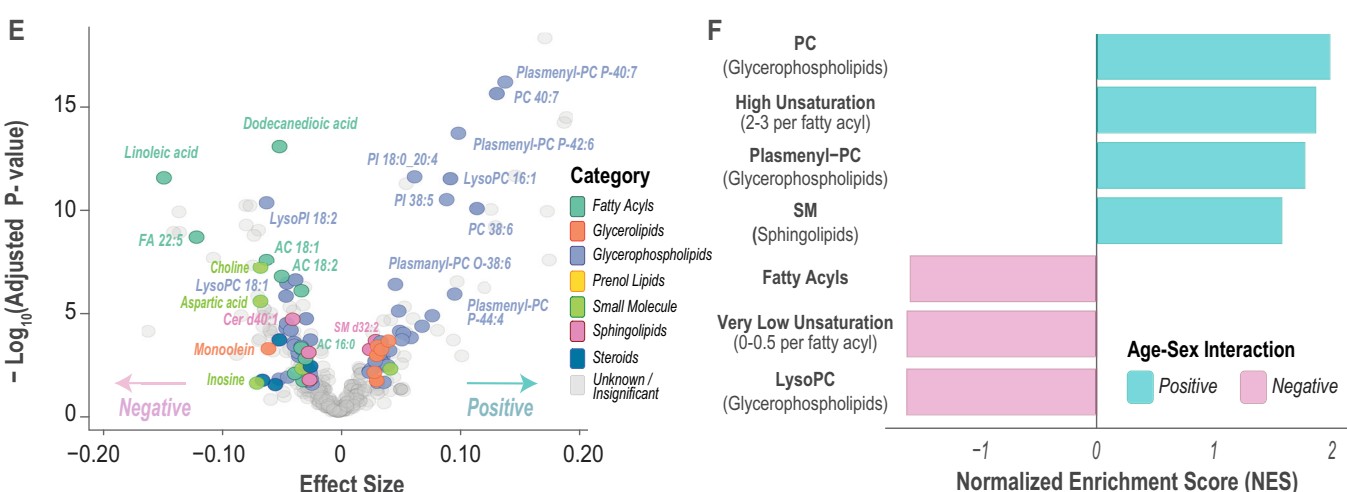

**Figure 6. Age-associated biomolecular profiles and their modulation by diet and sex.**

(A) Associations of biomolecules ($n = 359$) with age were determined using linear regression models. Statistical significance was evaluated using likelihood ratio tests ($x^2$ ANOVA), which included diet, age, and sex as predictors. Significant biomolecules (adjusted $p$ value <0.05, $n = 27$) are colored by category, while non-significant ($n = 332$) and unidentified features are shown in gray. (B) Rank-based enrichment was based on the linear regression results for aging, and the top enrichment categories are plotted as normalized enrichment scores (NES). (C) Associations of biomolecules ($n = 359$) with the age-by-diet interaction. Effect sizes are plotted against $-$log10 adjusted $p$ values, with significant biomolecules ($n = 15$) highlighted by category. (D) Rank-based enrichment was based on the linear regression results for age-diet interactions, and the top enrichment categories are plotted as normalized enrichment scores (NES). (E) Associations of biomolecules ($n = 359$) with the age-by-sex interaction. Effect sizes are plotted against $-$log10 adjusted $p$ values, with significant biomolecules ($n = 97$) highlighted by category. (F) Rank-based enrichment was based on the linear regression results for age–sex interactions, and the top enrichment categories are plotted as normalized enrichment scores (NES). Data information: For (A, C, E), statistical significance for the volcano plots was assessed using log likelihood ratio tests ($x^2$ ANOVA), and $p$ values were adjusted for multiple testing hypothesis using the Benjamini–Hochberg method. Source data are available online for this figure.

decreasing with age (Fig. 6A,B). This anti-correlation was stronger in Control-fed samples vs. the CR ones, potentially suggesting that CR modifies these age-associated changes. This is further evidenced by CR leading to the reduced rate of change of DGs with aging and an increased rate of change of lysoPCs and plasmenyl-PCs with age (Fig. 6C,D). This anti-correlation between lysoPCs and plasmenyl-PCs could be related to a recycling mechanism of plasmenyl-PCs that have a sequestered reactive oxygen species (ROS) at their vinyl ether linkage, later getting recycled into lysoPCs (Faria et al, 2024).

To further explore differences between CR and Control-fed monkeys, we calculated the difference between correlation values for each pair of biomolecules, e.g., CR Tau$_i$ – Control-fed Tau$_i$, where $i$ is the correlation between two biomolecules (Fig. 7B). Here, we found that the differences in correlation values follow a Gaussian distribution centered around 0, with only a small number of biomolecule pairs having a different correlation in CR monkeys vs. Controls. To explore the biomolecule pairs that differ between CR and Control-fed monkeys, we plot a chord diagram for the top differing pairs, based on their delta Tau (Fig. 7C). An example of a biomolecule pair that shows a positive delta Tau value is between SM d36:1 and lysoPE 20:4; these molecules have a negative correlation in Control monkeys (Tau = −0.35) but no strong correlation in CR monkeys (Fig. 7D). In contrast, an example of a negative delta Tau biomolecule pair is between monoolein and Cer[NS] d18:1_24:0 (Fig. 7E). In this pair, we see a positive correlation in Control monkeys (Tau = 0.38) and no correlation in CR monkeys. For both examples, these four biomolecules were significant with diet (Fig. 4 Source Data) and age (Fig. 6A), but surprisingly, they were not significant with the diet x age interaction. However, this might suggest these molecules undergo more subtle changes with CR, for example, differences in inflection points with age that have been observed in recent human studies (Shen et al, 2024).

## Discussion

In this study, we developed and validated a rapid metabolomic and lipidomic detection and quantitation method that is ideally suited for large-scale studies. Compared to previously established methods, our workflow has several critical advantages. First, this method is fast; a 15-min analysis time lowers instrument time and facilitates high-throughput data acquisition. Furthermore, the speed of the LC-MS/MS method is particularly valuable for analyzing time-sensitive samples where rapid and efficient analysis is crucial. Second, the BAMM extraction method captures both

polar metabolites and nonpolar lipids, minimizing the number of extractions needed (Muehlbauer et al, 2023). Third, the inclusion of a high-isopropanol mobile phase adapted from the multi-omics single-shot technology (nMOST) workflow (Kraus et al, 2025; He et al, 2021) allows for the elution of both polar metabolites and highly hydrophobic lipid species in a single run for efficient profiling of complex biological samples. The range and sensitivity of our method were confirmed using human plasma NIST1950, producing comparable data to methods that are far more labor-intensive and time-consuming. We leveraged this comprehensive and efficient method to analyze a large cohort of monkey plasma samples to measure the impact of CR on biomolecule profiles over time. The collection of 494 monkey plasma samples was analyzed in 10 days with an average of ~90 samples per day, including various quality control samples and blanks.

With this integrative method, we quantified 359 metabolite and lipid features in rhesus monkey plasma collected longitudinally across the adult lifespan. We found numerous biomolecules changed over the study, indicating that circulating features are responsive to age and that signatures of aging can be identified. These findings align with previous reports of metabolic reprogramming and lipid remodeling during the aging process (Hornburg et al, 2023; Darst et al, 2019) in the context of age-related diseases, including neurodegeneration (Hornburg et al, 2023; Choi et al, 2018; Dorninger et al, 2018), cardiovascular disease (Eichelmann et al, 2022), and diabetes (Beyene et al, 2021). Here, principal component analysis identified a strong effect of age on the plasma metabolite and lipid profile. Changes in metabolite and lipid abundance over the transition from early-adult to advanced-age life stages explained almost 30% of the variance across all variables in all individuals within the cohort.

A major goal of this study was to illuminate the changes in circulating molecules induced by CR. The beneficial effect of CR on mortality and morbidity in this study has been established, but how those phenotypic outcomes are reflected systemically at the molecular level has not been defined (Mattison et al, 2017). We identified a modest but consistent impact of CR, where the abundance of SM was higher in CR monkeys than in Controls, while DGs, TGs, and ceramides were lower. These outcomes could be viewed as a delay in aging, since aging-associated changes for each of these classes were identified in this study; notably, we found SMs were elevated in control-fed younger monkeys and declined with aging. These patterns are consistent with lipidomic and genetic studies in model organisms that demonstrate that remodeling of sphingolipids and glycerolipids is tightly coupled with lifespan regulation. In *C. elegans*, the loss of *asm-3* (acid

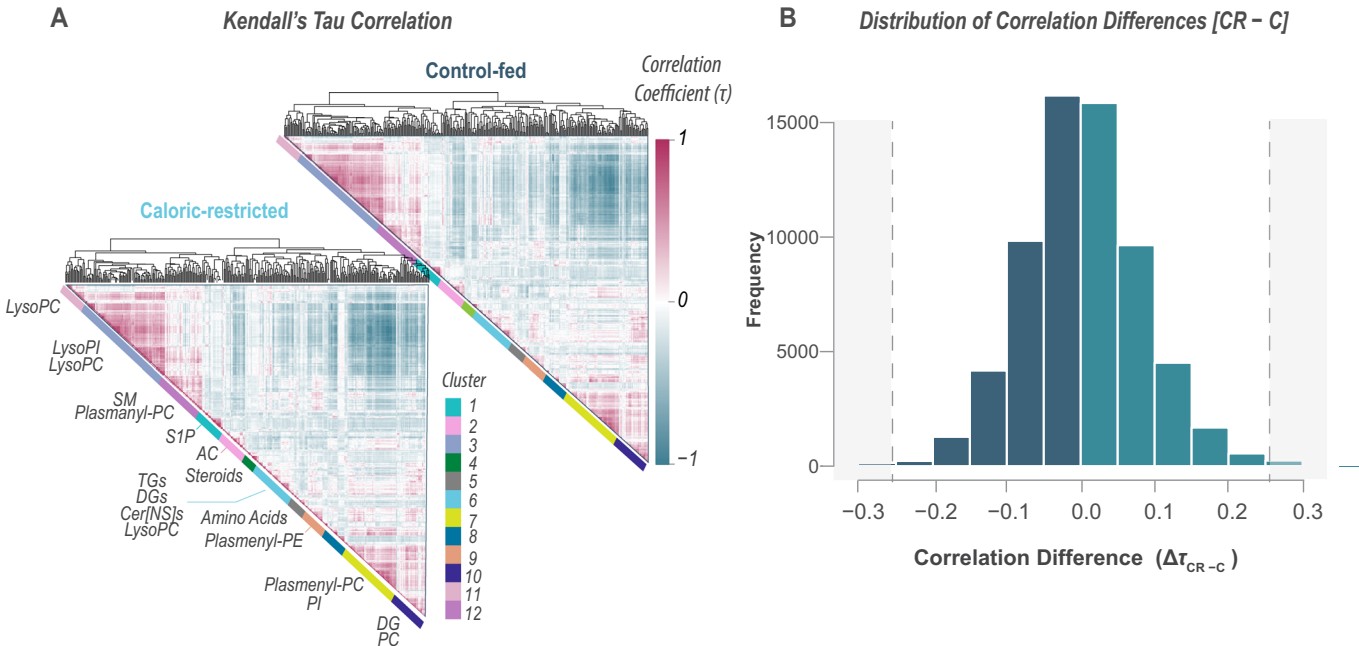

**A** *Kendall's Tau Correlation*

**B** *Distribution of Correlation Differences [CR − C]*

*Top Positive and Negative Delta Correlations Between Biomolecules*

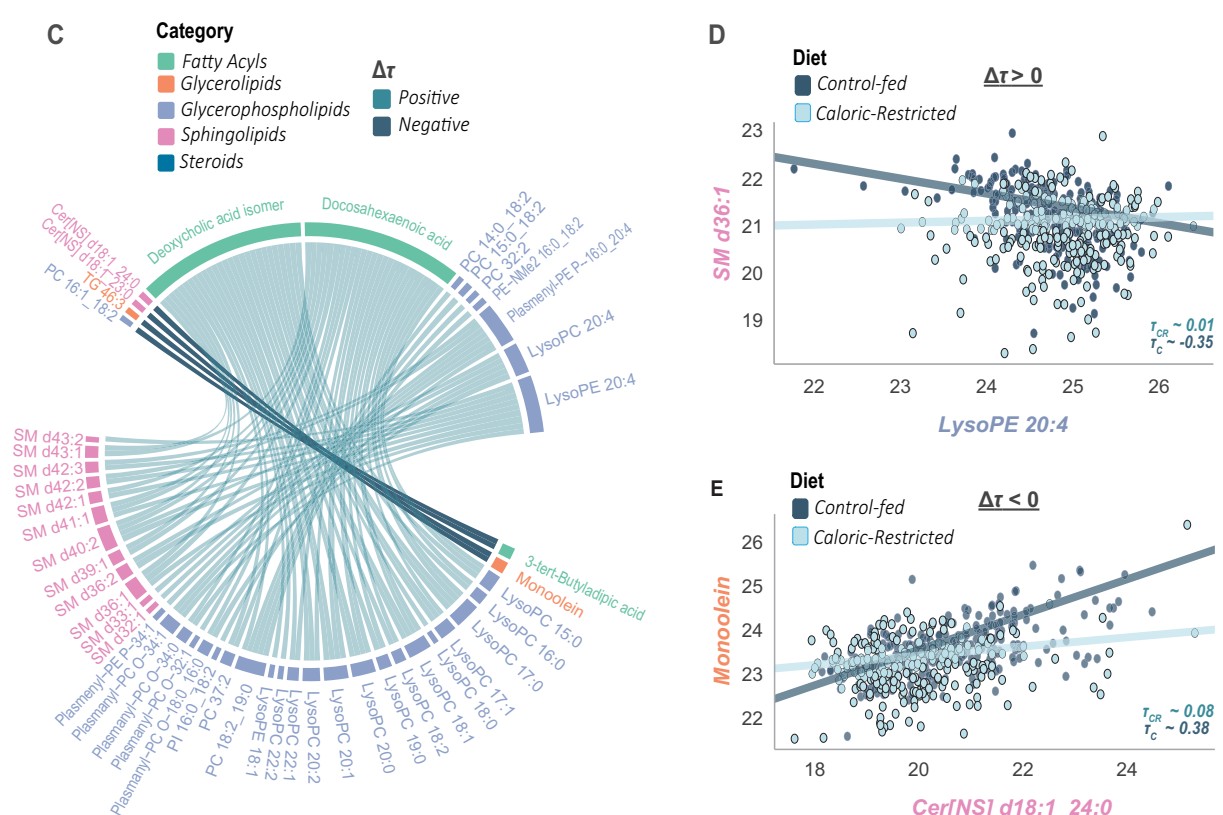

**Figure 7. Comparative analysis of correlation between metabolites and lipids in caloric-restricted (CR) and control-fed (C) monkeys.**

(A) Heatmaps represent Kendall's Tau correlation values between metabolites and lipids for CR and C groups, with hierarchical clustering based on the distance matrix of Controls and assigned into 12 clusters. (B) Histogram shows the distribution of the differences in correlation coefficients ($\Delta\tau = \tau_{CR} - \tau_C$) between the two diet groups. (C) The chord diagram highlights the strongest positive and negative delta correlations ($|\Delta\tau| \geq 0.26$). (D) An example of positive delta correlation (SM d36:1 *vs.* lysoPE 20:4). (E) a negative delta correlation (monoolein *vs.* cer[NS] d18:1_24:0) with the log2 abundances of these biomolecules are plotted for all samples; dots colored as indicated (C or CR). Data information: For (A), enriched categories were identified using Fisher's exact test and are shown next to the clusters.

sphingomyelinase), which converts sphingomyelin to ceramide, extended lifespan and promoted stress resistance by downregulating the insulin/IGF-1 signaling (IIS) pathway and activating the longevity-associated transcription factor DAF-16/FOXO (Kim and Sun, 2012). Notably, long-lived *asm-3* mutant worms also have reduced TGs abundance, consistent with our observations in the CR monkeys with aging (Staab et al, 2023). Together, these data suggest that the connection between disrupted sphingomyelin–ceramide balance and lifespan identified in short-lived species is conserved in primates.

Changes in sphingolipids and glycerolipids could also be a "signature" of CR and reflect a difference in the underlying lipid biology. In favor of this interpretation, our integrative analysis identified numerous pairs of molecules for which associations differed significantly between the two diet groups. The higher abundance of SM with CR aligns with evidence from human cohorts. Centenarians have higher SM levels than individuals 30 years their junior (Montoliu et al, 2014). At the tissue and cellular level, SMs influence membrane stability and signaling adaptations, contributing to differences in cellular function and response to external stimuli (Iqbal et al, 2017; Slotte, 2013). Sphingomyelins are key components of cell membranes, contributing to structural integrity and cellular signaling, although it is not clear how levels in plasma are regulated. Prior studies have linked sphingolipids to aging and several cellular hallmarks of aging (Tang et al, 2022; Li and Kim, 2021), where an imbalance of ceramides and sphingomyelins is thought to play a causal role in cellular dysfunction. Increases in plasma ceramides correlate with higher levels of cholesterol, neutral lipids, and fatty acids, suggesting a potential role as a risk factor for atherosclerosis (Ichi et al, 2006; Zhu and Scherer, 2024). Studies in humans and rodents link elevated hepatic and skeletal ceramides to insulin resistance (Hanamatsu et al, 2014; Kurek et al, 2015); yet, detailed sphingolipid profiles by acyl chain length in obesity-related diseases are poorly characterized. In addition to ceramides, higher abundances of DGs and TGs have been associated with loss of glycemic control, type II diabetes, and heart disease, independent of age (Zheng et al, 2019; Beyene et al, 2021). Moreover, increased abundance of TG is one of the diagnostic criteria for metabolic syndrome (Samson and Garber, 2014) and has clear links to metabolic disease (Gerl et al, 2019; Meikle and Summers, 2017). In this way, the lower levels of TG reported here for CR monkeys are suggestive of a healthier metabolic status. A growing body of literature links lipid metabolism to immune function and inflammation in the context of age-related diseases (Chiurchiù et al, 2022, 2018). Although speculative, it is possible that the age-related changes in abundance of circulating factors identified here are not just coincident with the known increase in risk for chronic disease, but may play a role in disease etiology, directly contributing to greater disease incidence. In this way, features that are impacted by CR, whether independent

of age or in a reversal of aging effects, are strong candidates in driving disease vulnerability.

We also examined sex-specific effects on circulating lipids and metabolites in male and female monkeys. A strong sex effect was detected in the levels of phosphocholine-containing lipids, with females having a higher abundance of these lipids than males. These findings align with previous literature from human and nonhuman primate studies (Yanai et al, 2024; Mohammadzadeh Honarvar et al, 2021; Darst et al, 2019), and lay the foundation for future integrative analyses. In this study, testosterone declined with age in both sexes, matching data from the Baltimore Longitudinal study, where levels were lower in both men and women of advanced age (Fabbri et al, 2016). In men, testosterone has been associated with increased risks of diabetes (Wittert et al, 2021), cardiovascular disease (Chen et al, 2024), and mortality (Yeap et al, 2021). It will be interesting to know if associations between age-related changes in circulating factors in humans and the incidence of age-related diseases are sex dimorphic and whether they are consistent throughout the lifespan or specific to a life stage. Translating discoveries from nonhuman primates to human aging would benefit from integrating plasma molecular profiling with functional readouts—such as tissue-specific metabolism, insulin sensitivity, and inflammatory status—in both species. This information would provide a more mechanistic understanding and delineate the pathways of aging. Ultimately, combining primate and human datasets in cross-species meta-analyses may help identify conserved, robust biomarkers of healthy aging and diet responsiveness, with broad implications for translational geroscience.

Overall, our findings highlight the impact of CR on lipid metabolism in the context of aging, with specific enrichment and depletion of lipid classes linked to membrane integrity, energy storage, and inflammatory pathways. By identifying key lipid classes and metabolites that are differentially regulated in CR monkeys over time, this study provides valuable insights into the molecular mechanisms underlying the healthspan-extending effects of CR. These results further demonstrate the power of MS-based omics approaches in uncovering biologically meaningful changes in complex metabolic networks, emphasizing their potential for future aging and metabolic health research. Although humans and monkeys differ in dietary history, environmental exposures, and metabolic rate, results from our monkey study consistently align well with data reported from the human clinical trial of CR known as CALERIE (Aversa et al, 2024; Huffman et al, 2022; Most and Redman, 2020), arguing that the findings presented here are translational and likely to be clinically relevant in health and age-related disease. These alignments strengthen the translational potential of our findings and suggest shared biological pathways that may underlie metabolic resilience in both species.

# Methods

## Reagents and tools table

| Reagent/resource | Reference or source | Identifier or catalog number |
| --- | --- | --- |
| **Experimental models** | | |
| Rhesus monkeys (*Macaca mulatta*) | Wisconsin National Primate Research Center (WNPRC) | |
| **Recombinant DNA** | | |
| **Antibodies** | | |
| **Oligonucleotides and other sequence-based reagents** | | |
| **Chemicals, enzymes and other reagents** | | |
| Acetonitrile (ACN; Optima™ LC/MS) | Thermo Fisher Scientific | A9554 |
| 2-Propanol (IPA; Optima™ LC/MS) | Thermo Fisher Scientific | A461-4 |
| Methanol (MeOH; Optima™ LC/MS) | Thermo Fisher Scientific | A454SK-4 |
| Toluene suitable for HPLC, 99.9% | Sigma-Aldrich | 34866-2 L |
| Ammonium formate (LC-MS Ultra) | Thermo Fisher Scientific | 6002012 |
| Formic acid (Pierce™) | Thermo Fisher Scientific | 28905 |
| n-Butanol (LiChrosolv®) | MilliporeSigma | 1019881000 |
| SPLASH LIPIDOMIX | Avanti Polar Lipids | 330707 |
| Cell Free $C^{13}$ $N^{15}$ Amino Acid Mixture | Sigma-Aldrich | 767964 |
| Human plasma standard reference material NIST1950 | Sigma-Aldrich | NIST1950 |
| Pooled, mixed-gender monkey serum | BioIVT | NHP02SRM-0104511 |
| **Software** | | |
| Compound Discoverer (CD) 3.3 | Thermo Fisher Scientific | |
| LipiDex 2 | https://github.com/coongroup/LipiDex-2 (Anderson et al, 2024b) | |
| **Other** | | |
| ACQUITY UPLC HSS T3 Column, 100 Å, 1.8 µm, 1 mm×100 mm | Waters | 186003536 |
| Vanquish UHPLC | Thermo Fisher Scientific | N/A |
| Orbitrap Eclipse Tribrid MS | Thermo Fisher Scientific | N/A |

## Experimental model and subjects' details

All animal procedures were performed at the Wisconsin National Primate Research Center (WNPRC) under approved protocols from the Institutional Animal Care and Use Committee of the Office of the Vice Chancellor for Research and Graduate Education of the University of Wisconsin–Madison. This study involved a retrospective analysis of previously collected and banked plasma samples from rhesus monkeys (*Macaca mulatta*) from a long-term Aging and Caloric Restriction (CR) study conducted at the WNPRC. The study included Indian-origin monkeys that were born and raised at the WNPRC, with all birthdates recorded. At the study's initiation, the animals were adults, aged 7–14 years, with no prior clinical or experimental background likely to affect disease risk or lifespan. The total enrollment of 76 monkeys included 46 males and 30 females. Further details on this monkey cohort can be found in previous publications (Colman et al, 2014, 2009; Rhoads et al, 2018). Briefly, animals were randomized into control or CR groups, matched based on baseline food intake, body weight, and age. Individualized food portions were determined using data collected over 3–6 months prior to the study commenced. For the CR group, food intake was then gradually reduced by 10% each month for three months until a total reduction of 30% was achieved. All animals were fed a semi-purified, nutritionally balanced, diet containing 15% protein and 10% fat. As part of the study design, animals with certain medical conditions received appropriate treatments.

## Purchased samples

Human plasma standard reference material 1950 was purchased from the National Institute of Standards and Technology (NIST). Pooled, mixed-gender monkey serum was purchased from BioIVT (rhesus macaques).

## Sample extraction

All chemicals are supplied in the Reagents and tools table. Plasma samples were assigned unique identifiers, and the analysts were blinded from the monkey identifiers and metadata until the completion of data collection and processing. Plasma metabolite and lipid extractions were performed in 12 batches. In each batch, for all metabolite/lipid extraction methods, study samples and two pooled quality control (QC) monkey serum samples were removed from −80 °C conditions and immediately placed on ice to thaw. All extraction solvents were chilled beforehand and of liquid chromatography (LC)-MS grade. All samples were rapidly extracted on dry ice once thawed. For each sample, 500 µL of chilled n-butanol:acetonitrile (ACN):water (3:1:1, v/v/v) was added to 10 µL of plasma in a 1.5 mL microcentrifuge tube. To monitor and correct for any possible variations, SPLASH LIPIDOMIX internal standard mixture (Avanti Polar Lipids, Inc.) and Cell Free $C^{13}$ $N^{15}$ Amino Acid Mixture (Sigma; diluted 1:100 from the stock) were added to the n-butanol:ACN:water extraction solvent before adding it to the samples. For evaluation of method sensitivity, we added dilutions of SPLASH mixture and Cell Free $C^{13}$ $N^{15}$ Amino Acid mixtures (using original stock concentrations, see Appendix Table S1) into equal volumes of NIST1950 plasma prior to extraction; dilutions evaluated were from 10E-4 to 1. Samples were then vortexed for 10 s and then centrifuged for 2 min at $14,000 \times g$ at 4 °C. 100 µL of the supernatant was then transferred to an amber glass autosampler vial with a fused glass insert. All extracts were dried by vacuum centrifugation for ~1 h. Samples were then reconstituted in 50 µL of methanol:water (1:1, v/v) and vortexed for 10 s. Vials were placed into an autosampler at 4 °C for analysis by LC-MS/MS. Before injection, samples were randomized within each batch in their queue for LC-MS/MS analysis to minimize potential bias. To further monitor instrument performance and correct for

within-batch effects, pooled QC monkey serum samples extracted alongside study samples in each batch were injected every four to six study samples.

## LC-MS/MS analysis

To perform chromatographic separations, a Vanquish Split Sampler HT autosampler (Thermo Scientific) was used to inject 2 μL of reconstituted extract onto a Waters ACQUITY HSS T3 C18 column (100 mm × 1 mm × 1.8 μm particle size) held at 40 °C throughout the analysis. Mobile phase A consisted of 0.2% formic acid in water. Mobile phase B consisted of 0.2% formic acid and 5 mM ammonium formate in isopropanol (IPA)/ACN (90:10, v/v). For the LC-MS/MS gradient (see Appendix Table S2), mobile phase B was initially held at 0% for 7 min at an 80 μL/min flow rate using a Vanquish Binary Pump (Thermo Scientific). The gradient then increased to 100% B for 4.25 min at the same flow rate, and it remained at 100% B with an increased flow rate of 100 μL/min for 0.75 min. After that, mobile phase B dropped to 0% (100 μL/min) for 2 min and the column was lastly re-equilibrated with mobile phase B at 0% for 1 min at an 80 μL/min flow rate before the next injection. The LC system was coupled to an Orbitrap Eclipse Tribrid mass spectrometer through a heated electrospray ionization (HESI II) source (Thermo Scientific). The source conditions were set as follows: aux gas flow rate at 5 units, sheath gas flow rate at 14 units, sweep gas flow rate at 2 units, temperature at 300 °C, spray voltage at |3.5 kV| for both positive and negative modes. The MS was operated in a polarity switching mode, acquiring full MS and MS/MS (Top2) spectra in both positive and negative ionization modes within the same injection with data-dependent Orbitrap MS2. Acquisition parameters for full MS scans in both modes were set as follows: 60,000 Orbitrap resolution; $1 \times 10^6$ automatic gain control (AGC) target; 50 ms ion accumulation time (max IT); 90–1350 $m/z$ scan range; and charge states were 1–2. MS/MS scans in both modes were then performed at 15,000 Orbitrap resolution; $1.25 \times 10^5$ AGC target; 22 ms max IT; 1.6 $m/z$ isolation window; fixed normalized collision energy (NCE) at 30%; and a dynamic exclusion of 30 s.

## Data processing

The resulting LC-MS data were processed using Compound Discoverer (CD) 3.3 (Thermo Scientific), LipiDex 2.0 (v. 0.1.10) (Anderson et al, 2024b), R (v. 4.4.0), and RStudio (v. 2024.04.0) (R Core Team, 2023).

For metabolomics/small molecules, raw data files were processed using CD 3.3 (Thermo Scientific). Peaks within a retention time (RT) range of 0 to 13 min and an MS1 precursor mass range of 0 to 5000 Da were selected and grouped into distinct compound groups based on a 10-ppm mass tolerance and a 0.4-min RT tolerance. Profiles that did not meet the following criteria were excluded from further processing: a minimum peak intensity of $1 \times 10^5$, a peak width of at least 3 min, a signal-to-noise (S/N) ratio of 1.5, and a fivefold intensity increase over blank samples. MS/MS spectra were then searched against the mzVault and mzCloud spectral libraries integrated within Compound Discoverer 3.3 for annotation assignment. The mzVault search included the following freely available and in-house generated libraries: "Bamba lab 34 lipid mediators library stepped NCE 10 30 45", "Bamba lab 598 polar

metabolites stepped NCE 10 30 45", "Customer_Library_ESI_HI-LIC_HCD_YZ_151entries", "Fiehn HILIC_3061entries, 'KI-GIAR_zic-HILIC_Pos_v0.90_814entries, "sixLibrariesDownloa-dedfromMoNA_2. For mzCloud, the "mzCloud_Offline_Autoprocessed" and "mzCloud_Offline_Reference" spectral libraries were used. Annotation parameters included a precursor mass tolerance of 15 ppm, a fragment mass tolerance of 10 ppm and 15 ppm (for mzVault and mzCloud, respectively), a retention time tolerance of 2 min, and an ion activation energy tolerance of 20 min. mzLogic scoring was then applied to assign annotations.

For lipidomics, raw data files were processed using CD 3.3 (Thermo Scientific) and LipiDex 2. The same workflow described for metabolomics was applied in CD 3.3. For annotation, using LipiDex 2, an in silico, in-house generated spectral library "LipiDex2_HCD_Formic" was used for MS/MS spectra searching. Lipid spectral matches with a dot product score greater than 500 and a reverse dot product score greater than 700 were then integrated into the compound discoverer results using the peak finder module in LipiDex 2. MS2 spectra were annotated at the molecular species level if the minimum spectral purity was at least 75%; otherwise, sum compositions were reported. Abbreviations are supplied in Appendix Table S3. Additional feature filtering was performed using the Degreaser module in LipiDex 2 to exclude adducts, dimers, in-source fragments, misidentified isotopes, and mismatched RTs. The filtering criteria included: (1) RT modeling with a maximum error of 0.4 min and at least three IDs per class for the model, (2) Peak quality factors (PQF) with a minimum peak rating of 4, and 3) linear dynamic range assessment to identify quantifiable features, requiring a minimum adjusted $R^2$ of 0.97 for features found in at least three consecutive samples in a dilution series of QC pooled monkey serum.

To further improve data quality, metabolite/lipid features were retained if they met two criteria: (1) they must be detected in at least 10% of the monkey study plasma samples ($n = 49$), and (2) either the RSD of the QC samples ≤ 85, or they must be detected in at least 20% of the QC samples ($n = 21$) to ensure that only high-quality, reliable features were kept. Following this, data normalization was performed by median normalization of the QC pooled monkey samples that were run with each batch to correct for within-batch effects. Subsequently, normalized features were further filtered for an RSD ≤30% in QC samples. Last, only $log2()$ transformed values of quantifiable features were scaled to have a mean of zero and kept for downstream biological analyses.

## Data analysis

### Principal component analysis

Using log2 transformed abundance measurements from monkey samples ($n = 494$), which included quantifiable lipid and small molecule features ($n = 359$), we performed principal component analysis (PCA) using the $prcomp()$ function in R (R version 4.4.0) (R Core Team, 2023).

## Determining significance with each covariable

To explore the relationship between each feature abundance and various factors of interest (Diet, age, and sex), the following multiple linear regression model was fit against the data using

*lmer*() function from *lme4* (Bates et al, 2015) package in R:

Relative Abundance ∼ Diet + Age + Sex + (1|Monkey ID)

To explore the combined effects of the predictors, we extended the model to include interaction terms:

Relative Abundance ∼ Diet + Age + Sex + (Age : Diet)
+ (Age : Sex) + (Diet : Sex) + (1|Monkey ID)

In these models, log2-normalized abundance values were used as a response variable, while diet, age, sex, and interaction terms were the predictor variables. Monkey ID was included as a random intercept to account for inter-individual variability within the study population, ensuring that differences between monkey individuals were properly accounted for in the analysis.

To assess the significance of each predictor, we used likelihood ratio tests (ANOVA chi-squared test, $\chi^2$), implemented in the *anova*() function in R, comparing each full model to a reduced model without the predictor of interest. This approach estimates the effect of each predictor while accounting for variation explained by all other predictors in the model. From each model, we obtained *p* values and the coefficient estimate (effect size), a measure of the magnitude of difference between variables that quantifies the impact of each predictor variable on the abundance of a given feature. To control the false discovery rate (FDR), *p* values were adjusted for multiple testing across all biomolecules using the *p.adjust*() function in R (method = BH), which applies the Benjamini–Hochberg procedure. All features were then evaluated for rank-based enrichment analysis.

### Enrichment analysis

Rank-based enrichment analysis was performed using the *fgsea* R package (Korotkevich et al, 2021). Categories were determined as metabolite/lipid class, lipids containing even/odd acyl chains, and degree of unsaturation as defined below. Lipids were categorized based on their average number of unsaturations per fatty acyl chain, with very low (0–0.5) and low (0.5–1) values representing primarily saturated or monounsaturated lipids, medium (1–2) values indicating a mix of monounsaturated and polyunsaturated lipids, and high (2–3) and very high (>3) values corresponding to increasing levels of polyunsaturation.

### Correlation analysis

The correlation between metabolites and diet types was calculated using the Kendall rank correlation method. The *cor.test()* function in R was utilized to conduct a correlation analysis between quantifiable metabolite and lipid abundances using Kendall's Tau (τ) method for each group of samples, CR vs control-fed, separately. This rank-based approach was chosen for its suitability with non-parametric data and its improved ability to manage tied ranks compared to other methods (i.e., Spearman's method) (Arndt et al, 1999). Kendall rank correlation matrices were computed separately for biomolecules in control-fed and CR monkey samples. Heatmaps were created to visualize the correlations within each sample set. To ensure consistency in clustering, hierarchical clustering with complete linkage was applied to reorder the columns and rows in the CR matrix based on the control matrix, using 1− cor(Biomolecule$_i$,Biomolecule$_j$) as the distance metric.

Mega-cluster groupings ($n = 12$) were then assigned manually, and the enriched biomolecule classes in each cluster were determined using the hypergeometric Fisher's exact test. Then, for each biomolecule pair, the difference between correlation values from each sample group (Δτ = τ$_{control}$ - τ$_{CR}$) was calculated to quantify shifts in biomolecular associations between the two sample sets. Biomolecule pairs with |Δτ| > 0.26 were retained for visualization. The selected pairs were then plotted using the *chordDiagram*() function from the *circlize* package in R (Gu et al, 2014).

For bar plots, data were shown as mean ± standard deviation or standard error of the mean as denoted in figure legends. Abundance values are presented as log2 (peak area) for metabolites and lipids. All graphs were created using the *ggplot2* package in R (Wickham, 2016) and further refined in Adobe Illustrator for additional aesthetics.

## Data availability

Further information and requests for resources should be directed to and will be fulfilled by the lead contacts, Katherine A. Overmyer (kovermyer@wisc.edu) and Rozalyn M. Anderson (rozalyn.anderson@wisc.edu). All code used to perform the computational analyses and reproduce the figures in this study is available on GitHub at https://github.com/sabouelhassan/Monkey_ MetaboLipidomics. Raw files are publicly available via MassIVE under accession number MSV000097353. Processed files are available on request.

The source data of this paper are collected in the following database record: biostudies:S-SCDT-10_1038-S44320-025-00177-3.

## Peer review information

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

## Acknowledgements

We are grateful for support from the NIH, including P41 GM108538, R01AG074503, R01AG040178, and the Simons Foundation. The WNPRC is supported by research facilities improvement programs, NCRR/ORIP grants P51RR000167/P51OD011106. This study was conducted using resources and facilities at the William S. Middleton Memorial Veterans Hospital, Madison, WI. The schematic illustrations in the synopsis figure were created at BioRender.com.

## Author contributions

**Salma I Abou Elhassan**: Conceptualization; Data curation; Formal analysis; Validation; Investigation; Visualization; Methodology; Writing—original draft; Writing—review and editing. **Josef P Clark**: Conceptualization; Investigation; Project administration; Writing—review and editing. **Di Kuang**: Validation; Writing—review and editing. **Timothy W Rhoads**: Conceptualization; Project administration; Writing—review and editing. **Ricki J Colman**: Conceptualization; Resources; Funding acquisition; Project administration; Writing—review and editing. **Joshua J Coon**: Resources; Supervision; Funding acquisition; Project administration; Writing—review and editing. **Rozalyn M Anderson**: Conceptualization; Resources; Supervision; Funding acquisition; Project administration; Writing—review and editing. **Katherine A Overmyer**: Conceptualization; Data curation; Formal analysis; Supervision; Funding acquisition; Validation; Investigation; Methodology; Writing—original draft; Project administration; Writing—review and editing.

Source data underlying figure panels in this paper may have individual authorship assigned. Where available, figure panel/source data authorship is listed in the following database record: biostudies:S-SCDT-10_1038-S44320-025-00177-3.

## Disclosure and competing interests statement

The authors declare the following competing financial interest (s): J.J.C. is a consultant for Thermo Fisher Scientific and Seer. J.J.C. is co-founder of CeleramAb, Inc.

# Expanded View Figures

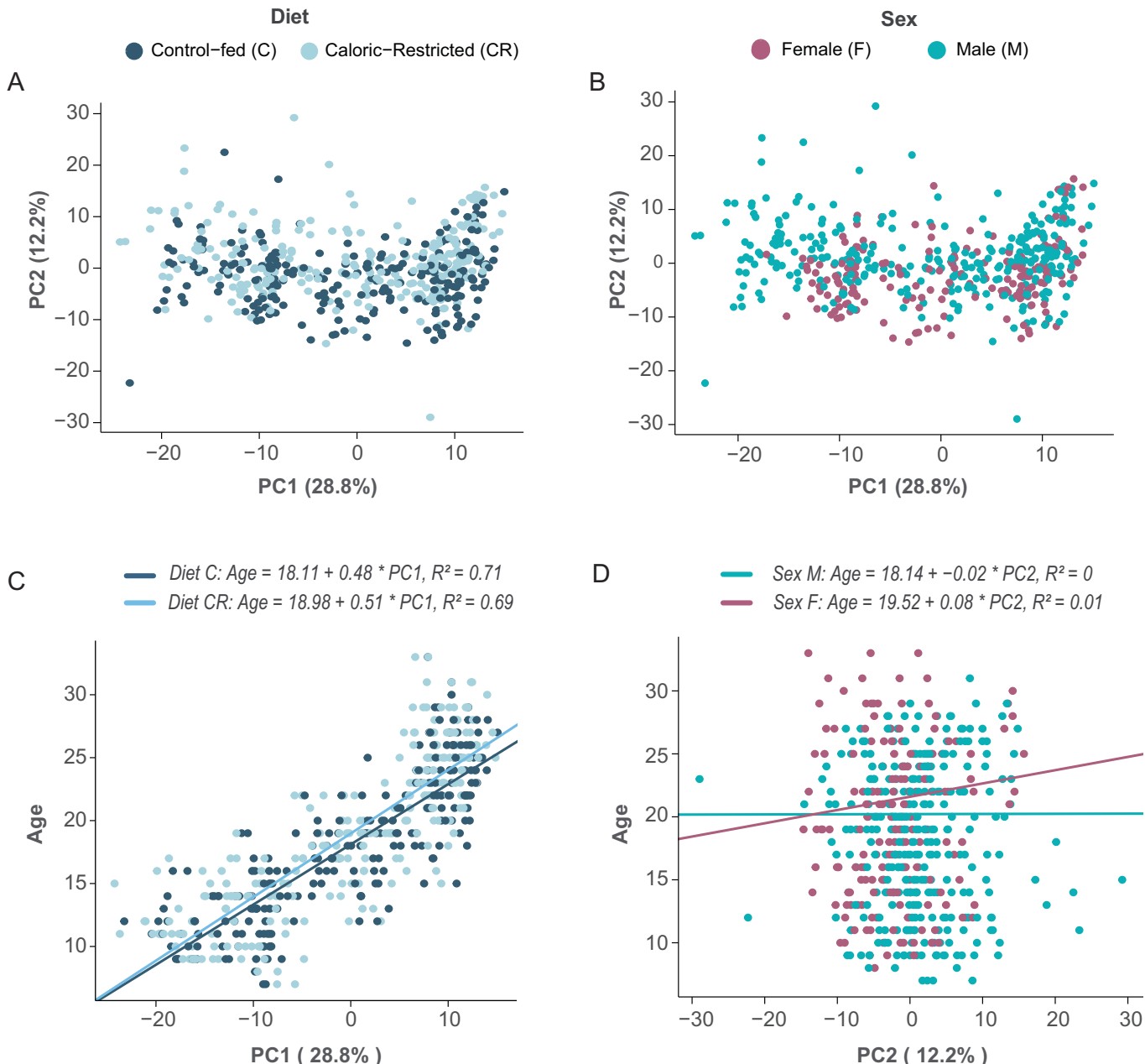

**Figure EV1. Evaluating the impact of diet and sex in the principal component analysis (PCA) of the metabolite data.**

(A, B) PCA of study samples using quantitative values from omics data (lipids and small molecules, $\log_2$ transformed and centered around 0 for $n = 494$ monkey plasma samples), shows that principal components 1 and 2 capture 28.8 and 12.2%, respectively, of the variance between monkey plasma samples; dots colored based on the type of diet applied to the monkeys (A) or sex of the studied monkeys (B). (C) The scatter plot shows principal component (PC) 1 relative to the age of monkeys to evaluate the effect of caloric restriction (CR) on PC1. (D) The scatter plot shows PC2 values relative to the age of monkeys to evaluate the effect of sex on PC2. Data information: Linear regression lines were fitted separately for each group in (C, D), with the coefficient of determination ($R^2$) calculated for each regression model. The regression equations, including slope and intercept, are displayed on each plot.

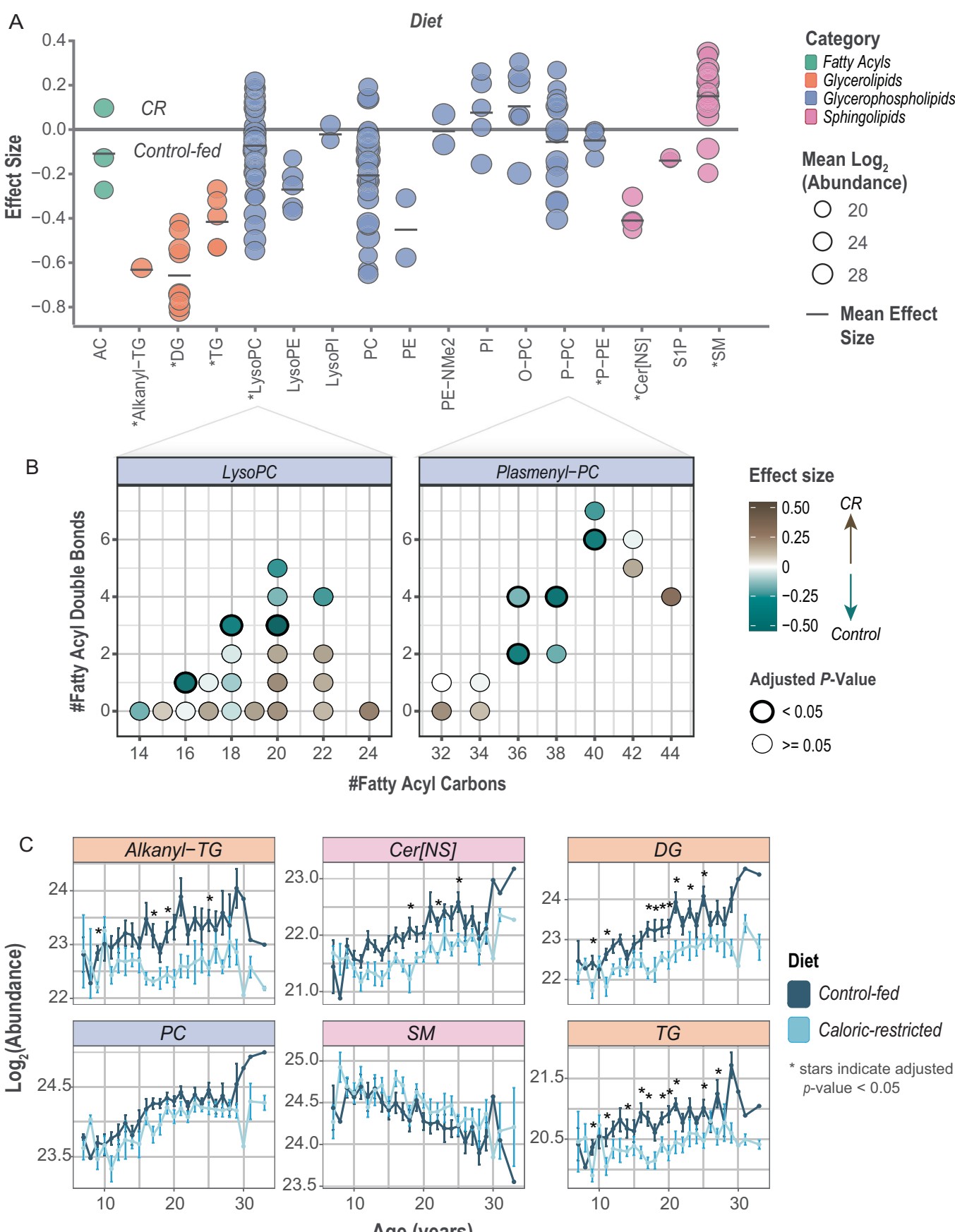

◄    **Figure EV2.  Dietary effects on lipid abundance in rhesus monkeys.**

(A) Effect sizes (CR vs. control-fed) are plotted for each lipid class and colored by lipid category. Each point represents a lipid species, with point size proportional to the mean $\log_2$ abundance of the related feature/lipid species across all samples. Horizontal bars indicate the mean effect size within each class. Negative values denote lower abundance under CR, while positive values denote higher abundance under CR. Lipid classes identified as significantly enriched or depleted by enrichment analysis are denoted with stars by name. (B) Detailed analysis of lysoPC and plasmenyl-PC are plotted by the fatty acyl chain length (number of fatty acyl carbons, x-axis) and the degree of unsaturation on fatty acyl chain (number of fatty acyl double bonds, y-axis). Circle stroke indicates the significance of lipid species based on their adjusted $p$ values from the regression models. Circle color represents effect size (CR vs. Control-fed). (C) Averaged abundances of lipid classes significantly associated with diet are plotted as a function of age; title colors are based on the lipid category, and dots show the mean abundance. Data information: For (C), error bars show the standard error of the mean, and stars indicate adjusted $p$ value $<0.05$ using Welch's unpaired $t$-test.

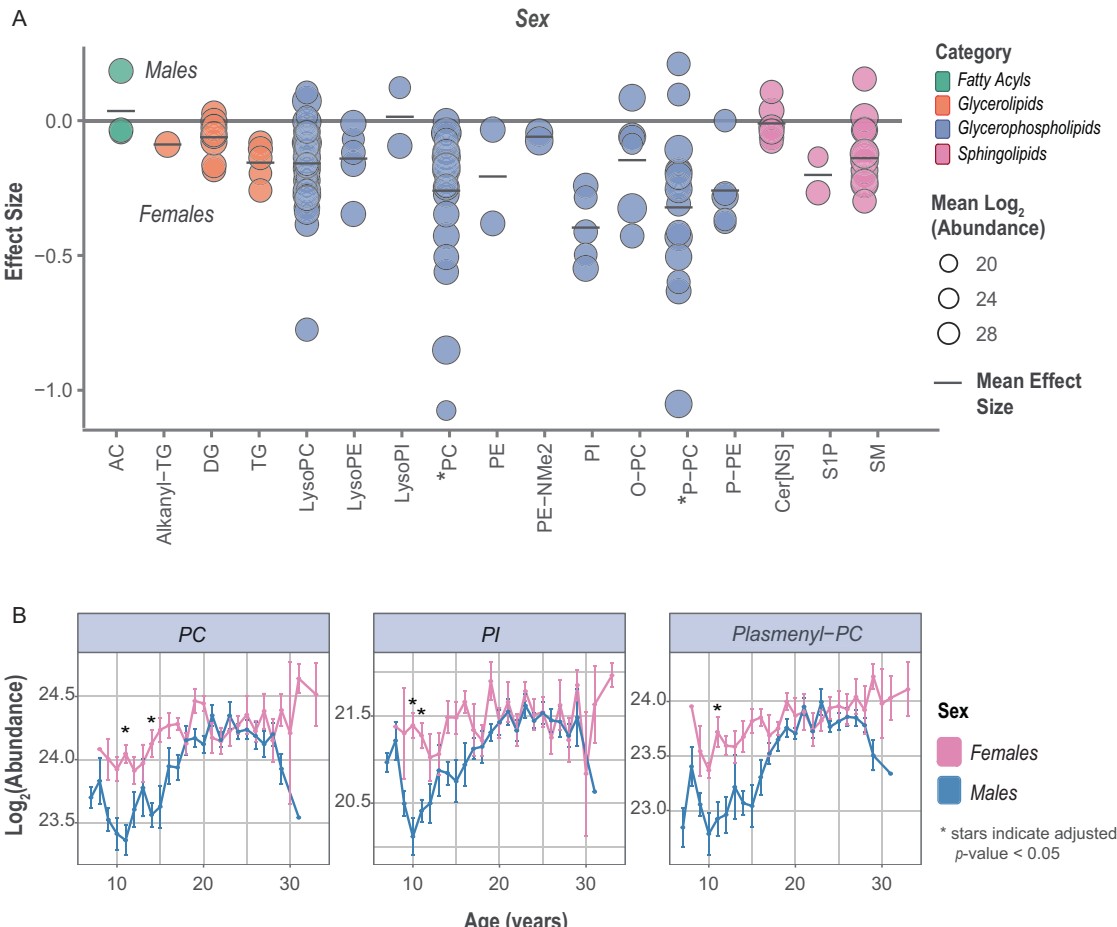

**Figure EV3. Sex-specific differences of lipid abundance in rhesus monkeys.**

(**A**) Effect sizes (males vs. females) across lipid classes are plotted and colored by lipid category. Each point represents a lipid species, with point size proportional to the mean log2 abundance of the related feature/lipid species across all samples. Horizontal bars indicate the mean effect size within each class. Negative values denote higher abundance in Females, while positive values denote higher abundance in Males. Lipid classes identified as significantly enriched or depleted by enrichment analysis are denoted with stars by name. (**B**) Averaged abundances of lipid classes significantly associated with sex are plotted as a function of age; title colors are based on the lipid category, dots show the mean abundance, and stars indicate adjusted *p* value <0.05. Data information: For (**B**), error bars show the standard error of the mean, and stars indicate adjusted *p* value <0.05 using Welch's unpaired *t*-test.

