## [Peer Review File · Molecular Systems Biology]

Aging-linked Systemic Lipid Signature is Reprogrammed by Caloric Restriction in Rhesus Monkeys

SALMA ABOUELHASSAN, Josef Clark, Di Kuang, Timothy Rhoads, Ricki Colman, Joshua Coon, Rozalyn Anderson, and Katherine Overmyer

Corresponding author(s): Katherine Overmyer (kovermyer@wisc.edu) , Rozalyn Anderson (rozalyn.anderson@wisc.edu), Katherine Overmyer (kovermyer@wisc.edu)

Review Timeline:

Submission Date:	8th May 25
Editorial Decision:	1st Jul 25
Revision Received:	6th Oct 25
Editorial Decision:	10th Nov 25
Revision Received:	24th Nov 25
Accepted:	25th Nov 25

Editor: Jingyi Hou

Transaction Report:

1st Jul 2025

Manuscript Number: MSB-2025-13100

Title: Aging Linked Systemic Lipid Signature is Reprogrammed by Caloric Restriction in Rhesus Monkeys

Author: SALMA ABOUELHASSAN

Josef Clark

Di Kuang

Timothy Rhoads

Ricki Colman

Joshua Coon

Rozalyn Anderson

Katherine Overmyer

Dear Dr. Overmyer,

Thank you for submitting your work to Molecular Systems Biology. First of all, I would like to apologise for the slow process. We have now heard back from two of the three reviewers who agreed to evaluate your manuscript. Unfortunately, after a series of reminders we did not manage to obtain a report from Reviewer #1. In the interest of time, and since the recommendations of the other two reviewers are quite similar, I prefer to make a decision now rather than further delaying the process. If we receive the comments from Reviewer #1, we will send them to you, and you can address the issues raised by Reviewer #1 together with those raised by the other two reviewers. You will see from the comments below that Reviewers #2 and #3 find the manuscript to be of interest. They raise, however, several important points, which should be convincingly addressed in a revision of this work.

I think that the recommendations of the reviewers are rather clear so there is no need to repeat the points listed below. All issues raised by the reviewers need to be satisfactorily addressed. As you may already know, our editorial policy allows in principle a single round of major revision so it is essential to provide responses to the reviewers' comments that are as complete as possible. Please feel free to contact me in case you would like to discuss in further detail any of the issues raised by the reviewers.

On a more editorial level, we would ask you to address the following issues:

- Please provide a .docx formatted version of the manuscript text (including legends for main figures, EV figures and tables). Please make sure that the changes are highlighted to be clearly visible.
- Please provide individual production quality figure files as .eps, .tif, .jpg (one file per figure).
- Please provide a .docx formatted letter INCLUDING the reviewers' reports and your detailed point-by-point responses to their comments. As part of the EMBO Press transparent editorial process, the point-by-point response is part of the Review Process File (RPF), which will be published alongside your paper.
- Please note that all corresponding authors are required to supply an ORCID ID for their name upon submission of a revised manuscript.
- We replaced Supplementary Information with Expanded View (EV) Figures and Tables that are collapsible/expandable online (see examples in <http://msb.embopress.org/content/11/6/812>). A maximum of 5 EV Figures can be typeset. EV Figures should be cited as 'Figure EV1, Figure EV2' etc... in the text and their respective legends should be included in the main text after the legends of regular figures.

Additional Tables/Datasets should be labeled and referred to as Table EV1, Dataset EV1, etc. Legends have to be provided in a separate tab in case of .xls files. Alternatively, the legend can be supplied as a separate text file (README) and zipped together with the Table/Dataset file.

For the figures and tables that you do NOT wish to display as Expanded View figures, they should be bundled together with their legends in a single PDF file called *Appendix*, which should start with a short Table of Content. Each legend should be below the corresponding Figure/Table in the Appendix. Appendix figures and tables should be referred to in the main text as: "Appendix Figure S1, Appendix Figure S2, Appendix Table S1" etc. See detailed instructions regarding expanded view here: <https://www.embopress.org/page/journal/17444292/authorguide#expandedview>.

- Before submitting your revision, primary datasets (and computer code, where appropriate) produced in this study need to be deposited in an appropriate public database (see <http://msb.embopress.org/authorguide> - dataavailability <https://www.embopress.org/page/journal/17444292/authorguide#dataavailability>).

The accession numbers and database should be listed in a formal "Data Availability" section (placed after Materials & Method) that follows the model below (see also <https://www.embopress.org/page/journal/17444292/authorguide#dataavailability>). Please note that the Data Availability Section is restricted to new primary data that are part of this study.

Data availability

- At EMBO Press we ask authors to provide source data for the main manuscript figures. You will receive a separate email with instructions for providing source data with your revised manuscript, including how to upload and organize the files. Additional information on source data and instruction on how to label the files are available

- Our journal encourages inclusion of *data citations in the reference list* to directly cite datasets that were re-used and obtained from public databases. Data citations in the article text are distinct from normal bibliographical citations and should directly link to the database records from which the data can be accessed. In the main text, data citations are formatted as follows: "Data ref: Smith et al, 2001". In the Reference list, data citations must be labeled with "[DATASET]". A data reference must provide the database name, accession number/identifiers and a resolvable link to the landing page from which the data can be accessed at the end of the reference. Further instructions are available at .

- We updated our journal's competing interests policy in January 2022 and request authors to consider both actual and perceived competing interests. Please review the policy <https://www.embopress.org/competing-interests> and update your competing interests if necessary.

Please use the heading "Disclosure statement and competing interests".

- All Materials and Methods need to be described in the main text using our 'Structured Methods' format. According to this format, the Methods section includes a Reagents and Tools Table (listing key reagents, experimental models, software and relevant equipment and including their sources and relevant identifiers) followed by a Methods and Protocols section describing the methods, ideally using a step-by-step protocol format. The aim is to facilitate adoption of the methodologies across labs.

Please download and fill our Reagents and Tools Table template (.docx), which you can find in our author guidelines: <https://www.embopress.org/page/journal/17444292/authorguide#structuredmethods>.

-Regarding data quantification:

Please ensure to specify the name of the statistical test used to generate error bars and P values, the number (n) of independent experiments (please specify technical or biological replicates) underlying each data point and the test used to calculate p-values in each figure legend. Discussion of statistical methodology can be reported in the materials and methods section, but figure legends should contain a basic description of n, P and the test applied.

Graphs must include a description of the bars and the error bars (s.d., s.e.m.).

- Please provide a "standfirst text" summarizing the study in one or two sentences (approximately 250 characters, including space), three to four "bullet points" highlighting the main findings and a "synopsis image" (550px width and 400-600 px height, PNG format) to highlight the paper on our homepage.

Here are a couple of examples:

<https://www.embopress.org/doi/10.15252/msb.20199356>

<https://www.embopress.org/doi/10.15252/msb.20209475>

<https://www.embopress.org/doi/10.15252/msb.209495>

When you resubmit your manuscript, please download our CHECKLIST (<https://www.embopress.org/pb-assets/embosite/EMBO%20Press%20Author%20Checklist-1642513524327.xlsx>) and include the completed form in your submission.

Please note that the Author Checklist will be published alongside the paper as part of the transparent process (<https://www.embopress.org/page/journal/17444292/authorguide#transparentprocess>).

If you feel you can satisfactorily deal with these points and those listed by the referees, you may wish to submit a revised version of your manuscript. Please attach a covering letter giving details of the way in which you have handled each of the points raised

by the referees. A revised manuscript will be once again subject to review and you probably understand that we can give you no guarantee at this stage that the eventual outcome will be favorable.

I look forward to receiving the revised manuscript soon.

Kind regards,
Jingyi

Jingyi Hou, PhD
Senior Editor
Molecular Systems Biology

We realize that it is difficult to revise to a specific deadline. In the interest of protecting the conceptual advance provided by the work, we recommend a revision within 3 months (29th Sep 2025). Please discuss the revision progress ahead of this time with the editor if you require more time to complete the revisions. Use the link below to submit your revision:

IMPORTANT: When you send your revision, we will require the following items:

1. the manuscript text in LaTeX, RTF or MS Word format
2. a letter with a detailed description of the changes made in response to the referees. Please specify clearly the exact places in the text (pages and paragraphs) where each change has been made in response to each specific comment given
3. three to four 'bullet points' highlighting the main findings of your study
4. a short 'blurb' text summarizing in two sentences the study (max. 250 characters)
5. a 'thumbnail image' (550px width and max 400px height, Illustrator, PowerPoint or jpeg format), which can be used as 'visual title' for the synopsis section of your paper.
6. Please include an author contributions statement after the Acknowledgements section (see <https://www.embopress.org/page/journal/17444292/authorguide>)
7. Please complete the CHECKLIST available at (<https://bit.ly/EMBOPressAuthorChecklist>). Please note that the Author Checklist will be published alongside the paper as part of the transparent process (<https://www.embopress.org/page/journal/17444292/authorguide#transparentprocess>).
8. When assembling figures, please refer to our figure preparation guideline in order to ensure proper formatting and readability in print as well as on screen:
<https://bit.ly/EMBOPressFigurePreparationGuideline>
See also figure legend guidelines: <https://www.embopress.org/page/journal/17444292/authorguide#figureformat>
9. Please note that corresponding authors are required to supply an ORCID ID for their name upon submission of a revised manuscript (EMBO Press signed a joint statement to encourage ORCID adoption). (<https://www.embopress.org/page/journal/17444292/authorguide#editorialprocess>)
Currently, our records indicate that the ORCID for your account is 0000-0002-1929-1229.

Link Not Available

11. Include a Reagents and Tools Table, which can be downloaded from our author guidelines (<https://www.embopress.org/page/journal/17444292/authorguide#structuredmethods>)

*** PLEASE NOTE *** As part of the EMBO Press transparent editorial process initiative (see our Editorial at <https://dx.doi.org/10.1038/msb.2010.72>), Molecular Systems Biology publishes online a Review Process File with each accepted manuscripts. This file will be published in conjunction with your paper and will include the anonymous referee reports, your point-by-point response and all pertinent correspondence relating to the manuscript. If you do NOT want this File to be published, please inform the editorial office at contact@molsystbiol.org within 14 days upon receipt of the present letter.

Reviewer #2:

In this study, the authors developed and validated a rapid, high-throughput metabolomic and lipidomic method optimized for large-scale biological studies. The workflow combines a 15-minute LC-MS/MS analysis with the BMM extraction technique, efficiently capturing both polar metabolites and nonpolar lipids in a single run. The method was applied to a large, longitudinal cohort of rhesus monkeys to investigate molecular signatures of aging and the impact of long-term caloric restriction (CR).

Using this approach, the authors analysed 494 plasma samples collected across the adult lifespan of monkeys. They identified 359 metabolite and lipid features, many of which exhibited significant age-related changes, reflecting the known metabolic reprogramming that occurs during aging. Notably, triglycerides (TGs) and diglycerides (DGs) increased with age, consistent with patterns observed in humans and their associations with metabolic diseases. Sphingomyelin (SM) levels, essential for membrane stability and cellular signaling, were higher in CR monkeys compared to controls, aligning with previous reports in human centenarians.

The study also revealed sex-specific differences in phosphocholine-containing lipids, with females showing higher abundance than males. Additionally, testosterone levels declined with age in both sexes, mirroring human aging trajectories.

This is a well-executed, technically innovative study that provides valuable insights into metabolic aging and CR in a highly relevant primate model. The methodological advancements and alignment with human findings strengthen its significance. With a more mechanistically driven, concise, and cautiously framed Results and Discussion, this manuscript would make a strong contribution to the fields of metabolomics, aging, and nutritional intervention research.

Comments:

1) While lipidomic changes are well-described, the manuscript lacks deeper mechanistic discussion linking specific lipids, amino acids and other metabolites to biological pathways of aging, insulin sensitivity, or disease prevention. The current version of the study is descriptive and does not provide any mechanistic explanation.

2) The study focuses on circulating plasma markers but does not integrate functional or tissue-level data (e.g., insulin sensitivity, liver/muscle metabolism) to validate the physiological relevance of the observed metabolic shifts.

3) Although sex differences are statistically demonstrated, the biological significance and potential hormonal or genetic contributors are not adequately discussed. The relationship between sex and age is not sufficiently discussed.

4) The interpretation of the correlation analysis is superficial. The delta correlation analysis shows altered biomolecule interactions under CR, but the functional implications or biological hypotheses arising from these changes are not explored.

5) While CR mitigates certain age-related lipid changes, PCA and regression analyses show that age remains the dominant factor. The narrative should avoid overstating the impact of CR relative to age effects.

6) The translational potential is implied, but more caution and discussion regarding species differences and applicability to human aging would strengthen the conclusion

7) Figure 2D illustrates a striking difference in the number of small molecules detected between the NIST1950 human plasma reference material (105 compounds) and the monkey serum samples (35 compounds), amounting to more than a twofold reduction in small molecule identification in the monkey samples. This substantial disparity raises important questions regarding the underlying causes. To what extent could biological differences between human plasma and monkey serum contribute to this result? Alternatively, might technical factors such as matrix effects, differences in metabolite concentrations, or potential limitations in database coverage for non-human primates play a role? Clarifying these factors is critical, especially given that cross-species comparisons are central to the study's broader translational relevance. The authors may wish to comment on these possibilities and, if applicable, outline whether future efforts to expand compound identification in monkey samples are planned

8) The data presented in Figure 3 demonstrate that age is the dominant factor driving variation in the metabolic and lipidomic profiles. However, the subsequent analyses comparing CR and control groups in Figure 4 appear to aggregate data across all ages without stratifying by age groups or life stages. This approach may obscure important age-dependent effects of CR on metabolism. It is well recognized that specific metabolites or lipid species may exhibit differences only during specific aging phases, which could be highly relevant for understanding the mechanisms of healthy aging and longevity. By not distinguishing between young and aging subjects, the analysis risks missing critical insights into how CR modulates metabolism dynamically over the lifespan. A stratified or interaction analysis incorporating age and diet would strengthen the study and potentially reveal biomarkers or pathways with age-specific responses to CR. The authors are encouraged to explore such age-diet interactions to fully capture the complexity of metabolic regulation in aging

Reviewer #3:

This manuscript presents a comprehensive longitudinal study investigating aging in rhesus monkeys and the effects of caloric restriction (CR) on delaying aging processes. Spanning nearly 25 years, the study tracks 76 monkeys from early adulthood to end-of-life, providing valuable insights into the long-term impacts of CR. In terms of technical innovation, Elhassan et al. have developed a combined omics approach that enables the detection of both polar metabolites and lipid species from a single specimen within a single LC-MS/MS analysis. This methodological advancement enhances efficiency and reduces sample variability. Conceptually, the study demonstrates that aging exerts the most profound influence on metabolite abundances, followed by sex and diet. Notably, the authors identify a distinct aging lipid signature, characterized by life-phase-specific alterations in sphingomyelins, diacylglycerols, and triacylglycerols. These findings contribute significantly to our understanding of metabolic changes associated with aging and dietary interventions.

While the study is robust and well-executed, the following points should be addressed to strengthen the manuscript before publication in *Molecular Systems Biology*:

1. The study spans nearly 25 years of aging in 76 monkeys with different diets. However, the article fails to mention any specific impact of caloric restriction (CR) on the lifespan of monkeys. It would be crucial to present these data to fully understand the effects of CR on longevity.
2. Figure 2D shows a marked detection disparity between the NIST1950 reference (105 small molecules) and monkey serum samples (45 small molecules). What factors account for this >2-fold difference in metabolite identification?
3. Figure 3D shows the strong influence of age as the primary driver of metabolic and lipidomic variation. However, the distinct differences in metabolic molecules and lipids observed across various ages, which could offer crucial insights for aging research, have unfortunately not been presented.
4. The data in Figure 3 underscore the dominant role of age as the primary driver of metabolic and lipidomic variation. However, the analyses of lipidomic and metabolomic differences between CR and control monkeys in Figure 4 did not distinguish between data from young and aging subjects. This approach risks overlooking critical information. For instance, components that exhibit differences exclusively during aging may hold even greater significance for aging research.
5. The authors discuss several types of changes in lipid components, yet Figure 4D fails to clearly classify and display these lipids with similar trends. This omission should be addressed through revision.
6. The relationship between sex dimorphism and aging is not adequately discussed. It is essential to provide relevant background information in the Introduction section.

Response to reviewers' comments

We thank all reviewers for taking the time to review our manuscript and for their thoughtful suggestions. Guided by those comments, we have revised the text to clarify and expand on the points raised by the reviewers, added new citations that frame the context for the work and the new insights derived, and conducted new analysis that is presented in the revised figures, including an entirely new figure 6. Specific comments are addressed below.

Remarks to the Author Reviewer #2:

In this study, the authors developed and validated a rapid, high-throughput metabolomic and lipidomic method optimized for large-scale biological studies. The workflow combines a 15-minute LC-MS/MS analysis with the BAMM extraction technique, efficiently capturing both polar metabolites and nonpolar lipids in a single run. The method was applied to a large, longitudinal cohort of rhesus monkeys to investigate molecular signatures of aging and the impact of long-term caloric restriction (CR).

Using this approach, the authors analysed 494 plasma samples collected across the adult lifespan of monkeys. They identified 359 metabolite and lipid features, many of which exhibited significant age-related changes, reflecting the known metabolic reprogramming that occurs during aging. Notably, triglycerides (TGs) and diglycerides (DGs) increased with age, consistent with patterns observed in humans and their associations with metabolic diseases. Sphingomyelin (SM) levels, essential for membrane stability and cellular signaling, were higher in CR monkeys compared to controls, aligning with previous reports in human centenarians.

The study also revealed sex-specific differences in phosphocholine-containing lipids, with females showing higher abundance than males. Additionally, testosterone levels declined with age in both sexes, mirroring human aging trajectories.

This is a well-executed, technically innovative study that provides valuable insights into metabolic aging and CR in a highly relevant primate model. The methodological advancements and alignment with human findings strengthen its significance. With a more mechanistically driven, concise, and cautiously framed Results and Discussion, this manuscript would make a strong contribution to the fields of metabolomics, aging, and nutritional intervention research.

1) While lipidomic changes are well-described, the manuscript lacks deeper mechanistic discussion linking specific lipids, amino acids and other metabolites to biological pathways of aging, insulin sensitivity, or disease prevention. The current version of the study is descriptive and does not provide any mechanistic explanation.

Response: We thank the reviewer for this comment, and we have substantially edited discussion points to better link our data with the existing literature and known effects of CR. A major feature of this study is the longitudinal nature of the data collection that contrasts with much of the rodent work that is cross-sectional. Another point of departure between this work and work in shorter-lived species is that the monkeys are aging and acquire human-equivalent geriatric conditions (Balasubramanian et al, 2017), but are not induced for specific diseases. In short-lived species, metabolites and lipids are usually interrogated against diet or genetically induced metabolic dysfunction or in transgenic models of specific diseases. In human studies, cohorts are often selected based on diagnosis of disease state or disease risk elevation where aging is an adjusted covariate and not the primary focus of investigation.

While we would benefit from deeper mechanistic studies, that is unfortunately outside of the scope of this manuscript. Instead, we have added more discussion points, making connections between our data and existing literature to help explain why these metabolites and lipids might be relevant to aging and overall health. We link our observations to aging (see response to comment #2) and insulin resistance.

Specifically for insulin resistance, we have referenced data on these monkeys, which show insulin sensitivity as one of the major outcomes of CR (Mattison *et al*, 2017).

“Interestingly, previous studies investigating circulating factors linked to insulin resistance reported a predictive model for loss of insulin sensitivity based on DG composition that was independent of adiposity and effective in advance of hyperglycemia (Polewski et al, 2015). Furthermore, in previously

published data on the monkey cohort presented in this study, fasting glucose levels in control monkeys begin to elevate around 23 yrs of age, indicating shifts towards insulin resistance (Mattison et al, 2017). Glucoregulatory impairment was also monitored by regular glucose tolerance tests in these monkeys, and control, but not CR, showed glucoregulatory impairment as early as 7 yrs of age. Notably, elevations in DGs and TGs appear earlier than elevations in fasting glucose (~ 20 yrs), and track closely with elevations in adiposity in the control vs. CR. Elevations in DGs and TGs frequently co-occur with insulin resistance (Ormazabal et al, 2018), and our data showing elevated DGs and TGs in control monkeys is in concordance with this observation.”

References:

Balasubramanian P, Howell PR & Anderson RM (2017) Aging and Caloric Restriction Research: A Biological Perspective With Translational Potential. *EBioMedicine* 21: 37–44

Mattison JA, Colman RJ, Beasley TM, Allison DB, Kemnitz JW, Roth GS, Ingram DK, Weindruch R, de Cabo R & Anderson RM (2017) Caloric restriction improves health and survival of rhesus monkeys. *Nat Commun* 8: 14063

2) The study focuses on circulating plasma markers but does not integrate functional or tissue-level data (e.g., insulin sensitivity, liver/muscle metabolism) to validate the physiological relevance of the observed metabolic shifts.

Response: We appreciate the reviewer’s feedback. We have chosen to focus this manuscript on circulating plasma metabolites rather than tissue-level data. Tissue-level changes in metabolites are not always reflected in plasma levels. However, to address this comment, we have added relevant citations linking our metabolite observations to functional studies throughout the manuscript (**see example provided below**). The improvements in health and survival of these specific animals have been reported previously and are cited in the paper. We provide for your consideration the longitudinal blood chemistry from these animals and point out that expected changes of lower TGs and lower cholesterol are evident in CR animals. Those data are not part of this current data set and are earmarked for a separate study that combines pathology with clinical indices of health in addition to health outcomes from the veterinary health records. For tissue-level data, we have companion papers at various stages of completion that investigate responses to age and to the CR diet in liver, skeletal muscle, adipose tissue, and heart. In aggregate, it is a large volume of data. We are not yet in a position to integrate among tissues, but that is certainly part of the larger plan. The current manuscript is narrower in focus but will be the foundation for the systems biology that we have planned.

The following text was added to the results section:

“Notably, this decline in LysoPCs with aging has been reported in previous studies on human subjects. Pan *et al.* (2023) reported that lysoPCs decline in the plasma of older Chinese adults, indicating their potential as aging biomarkers. Similarly, in the Baltimore Longitudinal Study of Aging, lower plasma lysoPC levels were associated with impaired mitochondrial oxidative capacity in skeletal muscles (Semba *et al.*, 2019).”

3) Although sex differences are statistically demonstrated, the biological significance and potential hormonal or genetic contributors are not adequately discussed. The relationship between sex and age is not sufficiently discussed.

Response: We appreciate the reviewer’s comment, particularly in light of recent advances that point to sex dimorphism in aging and agree that the interplay between sex and age is of interest. We have expanded the text to describe what is known about sex as a biological variable in the introduction. We have also conducted new analysis expanding upon our initial limited investigation of sex as a biological variable. We now show biomolecules that show significant age–sex interactions. The results are now presented in a **new Figure 6** and accompanying text. These additions can be found below (See **Results section**).

“We also evaluated how sex influenced biomolecules’ aging profiles. To do this, we used linear regression models and evaluated the significance of the interaction term between age and sex. Biomolecules’ age trajectories that are significantly influenced by sex are highlighted in **Figure 6E**. In this volcano plot, biomolecules with positive effect sizes indicate molecules that have a higher rate of change with age in males compared to females. An example of this is plasmenyl-PC P-40:7 that has a higher change with age in males, despite its lower abundance in males than in females (see also **Figure 5D**). Other biomolecules that have elevated change in males include PC 40:7, plasmenyl-PC P-42:6, and PC 38:6. These biomolecules with positive age-sex effect sizes were enriched in plasmenyl-PCs, PCs, and high unsaturated lipids (**Figure 6F**). In contrast, biomolecules that showed higher rates of change in females with age include linoleic acid, choline, aspartic acid, and acyl carnitines (ACs) 18:1, 18:2, and 16:0; these biomolecules were also enriched in fatty acyls, lysoPCs, and very low unsaturated lipids (**Figure 6E-F** and **Appendix Figure S6C**).”

Figure 6. Age-associated biomolecular profiles and their modulation by diet and sex. (A) Associations of biomolecules with age were determined using linear regression models that included diet, age, and sex as predictors. Significant biomolecules (adjusted p -value < 0.05) are colored by category, while non-significant and unidentified features are shown in gray. (B) Rank-based enrichment was based on the linear regression results for aging, and the top enrichment categories are plotted as normalized enrichment scores (NES). (C) Associations of biomolecules with the age-by-diet interaction. Effect sizes are plotted against $-\log_{10}$ adjusted p -values, with significant biomolecules highlighted by category. (D) Rank-based enrichment was based on the linear regression results for age-diet interactions, and the top enrichment categories are plotted as normalized enrichment scores (NES). (E) Associations of biomolecules with the age-by-sex interaction. (F) Rank-based enrichment was based on the linear regression results for age-sex interactions, and the top enrichment categories are plotted as normalized enrichment scores (NES). Data information: Statistical significance for the volcano plots was assessed with log likelihood-ratio tests, and p -values were adjusted using the Benjamini-Hochberg method.

We also made some additional edits to the text to explain potential hormonal contributions:

“the higher abundance of PCs and plasmeyl-PCs in females in early life could suggest a link to their estrogen status (Figure EV3B). Estrogen is known to regulate phospholipid metabolism, specifically by activating the phosphatidylethanolamine N-methyltransferase (PEMT) enzyme essential for PC synthesis (Resseguie et al, 2007). However, it is challenging to put these data into the context of menopause in this cohort given the current data. Overall, the observed sex dimorphism in lipid and metabolite profiles, including the higher abundance of testosterone in males and specific choline-containing phospholipids (e.g., PCs and plasmeyl-PCs) in females, strongly suggests a role for hormonal regulation in driving these metabolic differences.”

4) The interpretation of the correlation analysis is superficial. The delta correlation analysis shows altered biomolecule interactions under CR, but the functional implications or biological hypotheses arising from these changes are not explored.

Response: We acknowledge the reviewer’s point and have expanded our discussion of the correlation analysis. In the revised manuscript, we now describe these findings in greater detail and connect them to the results from our linear regression models (Figures 4–6) to provide biological context and potential interpretations. We added text to the **Results section** as underlined below:

“To explore the association among biomolecules as a function of diet, we performed a Kendall’s Tau (τ) correlation analysis on biomolecules in control-fed monkeys ($n=236$ samples) and in CR monkeys ($n=258$ samples). Correlation matrices were visualized as heatmaps for Control and CR monkeys separately, with hierarchical clustering of the Control matrix imposed on the CR matrix to enable comparison across correlations (Figure 7A). There was an overall similarity in the pattern of correlations between Control and CR samples, and biomolecules like SMs (cluster 12), lysophospholipids (clusters 3 and 11), and TGs and DGs (cluster 6) exhibited high within-class positive correlations. Notably, the lysophospholipid clusters, enriched in lysoPIs and lysoPCs, were strongly anti-correlated with clusters containing plasmeyl-PCs, PIs, DGs, and PCs (clusters 7 and 10). This anti-correlation is likely driven by age-induced changes. PCs were among the most significant biomolecules increasing with age, while lysoPCs were decreasing with age (Figure 6A-B). This anti-correlation was stronger in Control-fed samples vs. the CR ones, potentially suggesting that CR modifies these age-associated changes. This is further evidenced by CR leading to the reduced rate of change of DGs with aging and an increased rate of change of lysoPCs and plasmeyl-PCs with age (Figure 6C-D). This anti-correlation between lysoPCs and plasmeyl-PCs could be related to a recycling mechanism of plasmeyl-PCs that have a sequestered reactive oxygen species (ROS) at their vinyl ether linkage, later getting recycled into lysoPCs (Faria et al, 2024).”

“To explore the biomolecule pairs that differ between CR and Control-fed monkeys, we plot a chord diagram for the top differing pairs, based on their delta Tau (Figure 7C). An example of a biomolecule pair that shows a positive delta Tau value is between SM d36:1 and lysoPE 20:4; these molecules have a negative correlation in Control monkeys (Tau = -0.35) but no strong correlation in CR monkeys (Figure 7D). In contrast, an example of a negative delta Tau biomolecule pair is between monoolein and Cer[NS] d18:1_24:0 (Figure 7E). In this pair, we see a positive correlation in Control monkeys (Tau = 0.38) and no correlation in CR monkeys. For both examples, these four biomolecules were significant with diet (Figure 4 Source Data) and age (Figure 6A), but surprisingly, they were not significant with the diet x age interaction. However, this might suggest these molecules undergo more subtle changes with CR, for example, differences in inflection points with age that have been observed in recent human studies (Shen et al, 2024).”

5) While CR mitigates certain age-related lipid changes, PCA and regression analyses show that age remains the dominant factor. The narrative should avoid overstating the impact of CR relative to age effects.

Response: We certainly agree that age has the dominant effect on the circulating molecules identified here. We recognize that a fairer description in the narrative would be to emphasize that the CR-driven changes we observed are in the context of aging. The updated text, underlined in the manuscript, now reads:

“Overall, our findings highlight the impact of CR on lipid metabolism in the context of aging, with specific enrichment and depletion of lipid classes linked to membrane integrity, energy storage, and inflammatory pathways. By identifying key lipid classes and metabolites that are differentially regulated in CR monkeys over time, this study provides valuable insights into the molecular mechanisms underlying the healthspan-extending effects of CR.”

6) The translational potential is implied, but more caution and discussion regarding species differences and applicability to human aging would strengthen the conclusion.

Response: In the revised manuscript, we highlight key physiological and metabolic commonalities between rhesus macaques and humans, as well as differences that should be considered in translational interpretation. We have also clarified that while our results provide valuable insights into conserved biological processes, further validation in human cohorts is necessary to confirm their applicability.

First, we have added the following text in the **second paragraph of the Introduction** section, highlighting differences and applicability to human aging:

“Among nonhuman primates, rhesus macaques (*Macaca mulatta*) are a valuable model for aging due to their high genetic similarity to humans, sharing ~93% of their overall genome sequence and over 97% similarity in protein-coding genes and amino acid sequences (Gibbs *et al*, 2007). Not only do they share this genetic homology, but they also exhibit lipid and lipoprotein metabolism patterns that are highly comparable to those of humans, along with parallel trajectories of age-related physiological changes (Chiou *et al*, 2020; Stefanoni *et al*, 2020). These changes include declines in insulin sensitivity, sarcopenia, and immune system remodeling, making them an especially relevant and translatable model for studying human aging (Shang *et al*, 2017; Zheng *et al*, 2014; Pergande *et al*, 2024; Palliyaguru *et al*, 2021). Work involving human subjects is inherently complex due to multiple confounders that may directly or indirectly shape measured outcomes, such as lifestyle, diet composition, degree of adiposity, physical activity, alcohol consumption, substance abuse, and ethnicity. In addition to dietary and metabolic interventions, sex dimorphism is increasingly recognized as an important factor in aging biology. Across mammals, males and females frequently exhibit distinct lifespan trajectories, characterized by sex-specific differences in survival, disease susceptibility, and molecular aging signatures that impact the rate and nature of aging (Hägg & Jylhävä, 2021). These differences have been attributed to hormonal regulation, immune function, and metabolic pathways (Calabrò *et al*, 2023; Pomatto *et al*, 2018; Santos-Marcos *et al*, 2023). The importance of sex as a biological variable has gained traction in the biology of aging studies (Chen *et al*, 2022) and is expected to provide actionable insights of relevance to human aging. The complexity of human studies makes rhesus monkeys a good alternative to human subjects, and they are a highly translational model, with genomic, physiological, and behavioral similarities that are shared with humans and a spectrum of age-related diseases and conditions that mirror those prevalent in human aging (Balasubramanian *et al*, 2017b).”

We have also added the following text to **the last two paragraphs of the Discussion** section to address species-specific limitations and applicability to human aging research, as highlighted below:

“Translating discoveries from nonhuman primates to human aging would benefit from integrating plasma molecular profiling with functional readouts—such as tissue-specific metabolism, insulin sensitivity, and inflammatory status—in both species. This information would provide a more mechanistic understanding and delineate the pathways of aging. Ultimately, pooling primate and human

datasets in cross-species meta-analyses may help identify conserved, robust biomarkers of healthy aging and diet responsiveness, with broad implications for translational geroscience.

Overall, our findings highlight the impact of CR on lipid metabolism in the context of aging, with specific enrichment and depletion of lipid classes linked to membrane integrity, energy storage, and inflammatory pathways. By identifying key lipid classes and metabolites that are differentially regulated in CR monkeys over time, this study provides valuable insights into the molecular mechanisms underlying the healthspan-extending effects of CR. These results further demonstrate the power of MS-based omics approaches in uncovering biologically meaningful changes in complex metabolic networks, emphasizing their potential for future aging and metabolic health research. Although humans and monkeys differ in dietary history, environmental exposures, and metabolic rate, results from our monkey study consistently align well with data reported from the human clinical trial of CR known as CALERIE (Aversa *et al*, 2024; Huffman *et al*, 2022; Most & Redman, 2020), arguing that the findings presented here are translational and likely to be clinically relevant in health and age-related disease. These alignments strengthen the translational potential of our findings and suggest shared biological pathways that may underlie metabolic resilience in both species.”

7) Figure 2D illustrates a striking difference in the number of small molecules detected between the NIST1950 human plasma reference material (105 compounds) and the monkey serum samples (35 compounds), amounting to more than a twofold reduction in small molecule identification in the monkey samples. This substantial disparity raises important questions regarding the underlying causes. To what extent could biological differences between human plasma and monkey serum contribute to this result? Alternatively, might technical factors such as matrix effects, differences in metabolite concentrations, or potential limitations in database coverage for non-human primates play a role? Clarifying these factors is critical, especially given that cross-species comparisons are central to the study's broader translational relevance. The authors may wish to comment on these possibilities and, if applicable, outline whether future efforts to expand compound identification in monkey samples are planned.

Response: The reviewer is correct that matrix effects and variation in abundance could account for small molecule detection differences between NIST 1950 human plasma and monkey serum. However, we found that differences in the number and identity of detected small molecules between monkey serum and NIST1950 human plasma likely reflect exposome differences — the sum of environmental exposures, diet, and lifestyle factors. For example, molecules that were only found in humans include drug metabolites, plasticizers, pesticides, and sunscreen components. These exposome-related molecules are unlikely to be present in the monkey serum. These annotations were not subjected to extensive manual validation for isomer differentiation or filtering of putative biomolecules. To address this comment, we provided a supplementary table (Table EV1) outlining the metabolites’ “source” provided by the Human Metabolome Database (HMDB), which includes categories like exogenous, endogenous, or food sources. We have also added a short description of this analysis in the results section. See the underlined text below (**Results section**):

“We first assessed the number of features identified in triplicate of NIST1950 and pooled monkey serum samples using Compound Discoverer 3.3 (Thermo Fisher Scientific) and LipiDex2 (Anderson *et al*, 2024). These annotations were not subjected to extensive manual validation for isomer or putative biomolecules. For this work, we only considered Level 1 and Level 2 annotations (Schrimpe-Rutledge *et al*, 2016); Level 2 annotations were enabled by data-dependent acquisition (DDA) of MS2 spectra.”

“Surprisingly, we detected roughly twice as many small molecules in NIST1950 human plasma compared to monkey serum. Notably, many of the small molecules identified only in NIST1950 were consistent with exposome-related compounds, such as drug metabolites, plasticizers, pesticides, and personal care product components (i.e., sunscreen and cosmetics ingredients), and these molecules are not expected in the monkey serum. We summarized the HMDB-listed ‘sources’ and ‘roles’ of these small molecules in **Table EV1**.”

8) The data presented in Figure 3 demonstrate that age is the dominant factor driving variation in the metabolic and lipidomic profiles. However, the subsequent analyses comparing CR and control groups in Figure 4 appear to aggregate data across all ages without stratifying by age groups or life stages. This approach may obscure important age-dependent effects of CR on metabolism. It is well recognized that specific metabolites or lipid species may exhibit differences only during specific aging phases, which could be highly relevant for understanding the mechanisms of healthy aging and longevity. By not distinguishing between young and aging subjects, the analysis risks missing critical insights into how CR modulates metabolism dynamically over the lifespan. A stratified or interaction analysis incorporating age and diet would strengthen the study and potentially reveal biomarkers or pathways with age-specific responses to CR. The authors are encouraged to explore such age-diet interactions to fully capture the complexity of metabolic regulation in aging.

Response: We appreciate this important point. Age is indeed the dominant driver of variance (see **Figure 3**), and we agree that age-specific effects of caloric restriction could be important for understanding the underlying mechanism. For the analyses presented in **Figure 4**, diet effects were estimated from linear regression models that included age and sex as covariates, and thus the observed diet effects are adjusted for any differences in age distribution or age-related variation. We recognize that this may have not been obvious from our initial description of the text to readers. This model assessed the relationship between each quantified feature intensity and three key factors: Diet (CR vs. Control), Sex, and Age. With this approach, we were able to evaluate the effect of CR in the context of aging. In addition, we have added a dedicated analysis of age main effects and age-diet interactions in the new Figure 6. We now report that 193 biomolecules show significant associations with age, and that 20 biomolecules exhibit significant age-diet interactions, demonstrating age-contingent CR effects. We acknowledge the possibility of non-linear effects of age, suggested by these new data, and are conducting further in-depth analysis to investigate this for a follow-up paper.

See the text added to the following sections and new Figure 6 below:

(A) In the Methods section, we further explained the linear regression models as highlighted in the underlined text below:

“To explore the relationship between each feature abundance and various factors of interest (Diet, Age, and Sex), the following multiple linear regression model was fit against the data using *lmer()* function from *lme4* (Bates *et al*, 2015) package in R:

Relative Abundance ~ Diet + Age + Sex + (1 | Monkey ID)

To explore the combined effects of the predictors, we extended the model to include interaction terms:

Relative Abundance ~ Diet + Age + Sex + (Age:Diet) + (Age:Sex) + (Diet:Sex) + (1 | Monkey ID)

To assess the significance of each predictor, we used likelihood ratio tests (ANOVA chi-squared test, χ^2), implemented in the *anova()* function in R, comparing each full model to a reduced model without the predictor of interest. This approach estimates the effect of each predictor while accounting for the variation explained by all other predictors in the model. From each model, we obtained *p*-values and the coefficient estimate (effect size), a measure of the magnitude of difference between variables that quantifies the impact of each predictor variable on the abundance of a given feature.”

(B) In the Results section, we clarified the model in the text below:

“This model assessed the relationship between each quantified feature intensity and three key factors: Diet (CR vs. Control), Sex, and Age. With this approach, we were able to evaluate the effect of CR in the context of aging, or said another way, at any given age we are evaluating the effect of CR.”

and (C) in the new Figure 6, we added a dedicated analysis of age main effects and age-diet interactions in the text below:

“Due to the strong influence of age on the biomolecule abundance, we assessed both the impact of age and the interactions between age, diet, and sex using the linear regression approach. We found 312 biomolecules significant with age (**Figure 6A**). The biomolecules most increasing with age include LysoPC 20:5, azelic acid, and PI 18:0_20:4. These biomolecules elevated with age were enriched in phosphatidylcholines and high unsaturated lipids (**Figure 6B** and **Appendix Figure S6A**). Biomolecules that were decreasing with age include lysoPI 18:0, aspartic acid, and suberic acid. Overall, these biomolecules that were reduced with age were enriched in sphingomyelins, lysoPC, and very low unsaturated lipids. Notably, this decline in LysoPCs with aging has been reported in previous studies on human subjects. Pan *et al.* (2023) reported that lysoPCs decline in the plasma of older Chinese adults, indicating their potential as aging biomarkers. Similarly, in the Baltimore Longitudinal Study of Aging, lower plasma lysoPC levels were associated with impaired mitochondrial oxidative capacity in skeletal muscles (Semba *et al.*, 2019).

As shown in **Figure 4D**, many significant molecules with diet also had changes with age. **Figure 6C** highlights 20 biomolecules with a significant age-diet interaction; these include DGs, which have a significantly higher change with aging in control-fed monkeys relative to the CR monkeys. These are in contrast to lysoPCs (19:0, 20:0, and 24:0), which are increasing more in CR monkeys with age compared to control-fed monkeys. This interaction term can be interpreted as differences in slope between control-fed and CR monkeys with respect to age. The biomolecules that had higher slopes in CR monkeys with age were enriched in sphingomyelins, lysoPCs, and low and very low unsaturated lipids (**Figure 6D** and **Appendix Figure S6B**). While molecules with reduced slopes in CR monkeys with age were enriched in DGs and Alkanyl-TGs.

Figure 6. Age-associated biomolecular profiles and their modulation by diet and sex. (A) Associations of biomolecules with age were determined using linear regression models that included diet, age, and sex as predictors. Significant biomolecules (adjusted p -value < 0.05) are colored by category, while non-significant and unidentified features are shown in gray. (B) Rank-based enrichment was based on the linear regression results for aging, and the top enrichment categories are plotted as normalized enrichment scores (NES). (C) Associations of biomolecules with the age-by-diet interaction. Effect sizes are plotted against $-\log_{10}$ adjusted p -values, with significant biomolecules highlighted by category. (D) Rank-based enrichment was based on the linear regression results for age-diet interactions, and the top enrichment categories are plotted as normalized enrichment scores (NES). (E) Associations of biomolecules with the age-by-sex interaction. (F) Rank-based enrichment was based on the linear regression results for age-sex interactions, and the top enrichment categories are plotted as normalized enrichment scores (NES). Data information: Statistical significance for the volcano plots was assessed with log likelihood-ratio tests, and p -values were adjusted using the Benjamini–Hochberg method.

Remarks to the Author Reviewer #3:

This manuscript presents a comprehensive longitudinal study investigating aging in rhesus monkeys and the effects of caloric restriction (CR) on delaying aging processes. Spanning nearly 25 years, the study tracks 76 monkeys from early adulthood to end-of-life, providing valuable insights into the long-term impacts of CR. In terms of technical innovation, Elhassan et al. have developed a combined omics approach that enables the detection of both polar metabolites and lipid species from a single specimen within a single LC-MS/MS analysis. This methodological advancement enhances efficiency and reduces sample variability. Conceptually, the study demonstrates that aging exerts the most profound influence on metabolite abundances, followed by sex and diet. Notably, the authors identify a distinct aging lipid signature, characterized by life-phase-specific alterations in sphingomyelins, diacylglycerols, and triacylglycerols. These findings contribute significantly to our understanding of metabolic changes associated with aging and dietary interventions.

*While the study is robust and well-executed, the following points should be addressed to strengthen the manuscript before publication in *Molecular Systems Biology*:*

1) *The study spans nearly 25 years of aging in 76 monkeys with different diets. However, the article fails to mention any specific impact of caloric restriction (CR) on the lifespan of monkeys. It would be crucial to present these data to fully understand the effects of CR on longevity.*

Response: We appreciate the reviewer's suggestion. Health and survival outcomes for the University of Wisconsin (UW) caloric-restricted monkey cohort have been reported previously (Mattison et al., 2017). However, we agree that the link between the monkey cohort and this paper was not clear to readers. To provide a clearer context for our multi-omics analysis of this monkey cohort, we have added a brief description of study outcomes, including the impact of CR on the lifespan of these monkeys from the previously published UW cohort data in Mattison et al. (2017). The text is added to the last paragraph of the Introduction as underlined below:

“Over the course of the Wisconsin Aging and Calorie Restriction Study, the hazard ratio (HR 1.9) indicated that at any time-point, the control monkeys had almost twice the rate of death when compared to CR animals, and that age-related conditions occurred at more than twice the rate in control animals compared to CR (HR 2.7) (Mattison et al, 2017).”

References:

Mattison JA, Colman RJ, Beasley TM, Allison DB, Kemnitz JW, Roth GS, Ingram DK, Weindruch R, de Cabo R & Anderson RM (2017) Caloric restriction improves health and survival of rhesus monkeys. *Nat Commun* 8: 14063

2) *Figure 2D shows a marked detection disparity between the NIST1950 reference (105 small molecules) and monkey serum samples (45 small molecules). What factors account for this >2-fold difference in metabolite identification?*

Response: We thank the reviewer for raising this point, which was also mentioned by Reviewer 2. Several factors could contribute to the observed difference in small molecule identifications between NIST 1950 human plasma and monkey serum. Technical factors such as matrix effects and differences in compound abundance between the two biological matrices could influence MS/MS fragmentation quality and thus the confidence of annotations.

However, we also believe that biological factors play a significant role. Specifically, many of the small molecules detected only in NIST 1950 human plasma are consistent with exposome-related compounds—the sum of environmental exposures, diet, and lifestyle factors—which are not expected in monkeys. Examples include drug metabolites, plasticizers, pesticides, and personal care product components such as sunscreen and cosmetics ingredients.

To provide a clearer context, we have now included a supplementary table (**Table EV1**) listing the Human Metabolome Database (HMDB) “source” and “role” annotations for all identified small molecules, allowing

readers to distinguish between endogenous and exogenous (i.e., drugs or food-derived compounds). This table highlights that many of the human-only identifications fall into the exogenous category. We have also clarified this point in the Results section and added a short description of this analysis in the results section. See the underlined text below:

“We first assessed the number of features identified in triplicate of NIST1950 and pooled monkey serum samples using Compound Discoverer 3.3 (Thermo Fisher Scientific) and LipiDex2 (Anderson *et al*, 2024). These annotations were not subjected to extensive manual validation for isomer or putative biomolecules. For this work, we only considered Level 1 and Level 2 annotations (Schrimpe-Rutledge *et al*, 2016); Level 2 annotations were enabled by data-dependent acquisition (DDA) of MS2 spectra.”

“Surprisingly, we detected roughly twice as many small molecules in NIST1950 human plasma compared to monkey serum. Notably, many of the small molecules identified only in NIST1950 were consistent with exposome-related compounds, such as drug metabolites, plasticizers, pesticides, and personal care product components (i.e., sunscreen and cosmetics ingredients), and these molecules are not expected in the monkey serum. We summarized the HMDB-listed ‘sources’ and ‘roles’ of these small molecules in **Table EV1**.”

3) *Figure 3D shows the strong influence of age as the primary driver of metabolic and lipidomic variation. However, the distinct differences in metabolic molecules and lipids observed across various ages, which could offer crucial insights for aging research, have unfortunately not been presented.*

Response: We thank the reviewer for this observation that was also articulated by Reviewer 2. In response, we have now included additional analyses highlighting the specific metabolic and lipidomic differences observed over age. These results are presented in the revised manuscript and illustrated in the new **Figure 6**, which provides a more detailed view of age-associated molecular changes and their potential relevance to aging research. See response to Reviewer 2 Comment 8 above.

4) *The data in Figure 3 underscore the dominant role of age as the primary driver of metabolic and lipidomic variation. However, the analyses of lipidomic and metabolomic differences between CR and control monkeys in Figure 4 did not distinguish between data from young and aging subjects. This approach risks overlooking critical information. For instance, components that exhibit differences exclusively during aging may hold even greater significance for aging research.*

Response: This is an excellent point. We have clarified in the main text that caloric restriction vs. control-fed analysis accounts for aging in the linear regression model. That is, at any given age, we are evaluating the effect of CR. We have also added the analyses for age and age-diet interaction in the new **Figure 6**.

5) *The authors discuss several types of changes in lipid components, yet Figure 4D fails to clearly classify and display these lipids with similar trends. This omission should be addressed through revision.*

Response: We have added a plot summarizing the effect of diet by lipid class in **Figure EV2**. We have also added similar plots for the effect of sex, age, and age interactions with diet and sex in **Figures EV3** and **Appendix Figure S6**. Additionally, we have added clarity to **Figure 4D** to explain why these biomolecules were selected for display (e.g., top 5 significant biomolecules). See revised **Figure 4**, and new **Figures EV2**, **EV3** and **Appendix Figure S6** below:

Figure 4. Lipidomic and metabolomic differences between caloric-restricted (CR) and control-fed (C) monkeys. (A) Associations of biomolecules with diet type (CR vs. Control) were determined using linear regression log likelihood-ratio tests, and p -values were adjusted using the Benjamini-Hochberg method. The $-\log_{10}$ adjusted p -values are plotted against the effect size; biomolecules with an adjusted p -value < 0.05 are colored based on their category, with non-significant and unidentified features shown in gray. (B) Rank-based enrichment plot based on the linear regression analysis shows the top enrichment categories with normalized enrichment scores (NES). (C) Comparison of biomolecule abundance among all CR and C samples highlights significantly differentially abundant biomolecules. (D) Biomolecule abundances of lipids significantly associated with diet are plotted as a function of age; title colors are based on the biomolecules category. Data information: For (C) ** and **** indicate $p < 0.01$ and $p < 0.0001$, respectively as calculated with ANOVA. For each boxplot, the middle horizontal line is the median, box margins are first and third quartiles, with vertical lines extending ± 1.5 -times the interquartile range, and each dot represents an individual sample. For (D) dots show the mean abundance, error bars show the standard error of the mean, and stars indicate p -value < 0.05 using Student's unpaired t -test.

Figure EV2. Dietary effects on lipid abundance in rhesus monkeys. (A) Effect sizes (CR vs. control-fed) are plotted for each lipid class and colored by lipid category. Each point represents a lipid species, with point size proportional to mean log₂ abundance of the related feature/lipid species across all samples. Horizontal bars indicate mean effect size within each class. Negative values denote lower abundance under CR, while positive values denote higher abundance under CR. Lipid classes identified as significantly enriched or depleted by enrichment analysis are denoted with stars by name. (B) Detailed analysis of lysoPC and plasmeyl-PC are plotted by the fatty acyl chain length (number of fatty acyl carbons, x-axis) and the degree of unsaturation on fatty acyl chain (number of fatty acyl double bonds, y-axis). Circle stroke indicates the significance of lipid species based on their adjusted *p*-values from the regression models. Circle color represents effect size (CR vs. Control-fed). (C) Averaged abundances of lipid classes significantly associated with diet are plotted as a function of age; title colors are based on the lipid category, dots show the mean abundance, error bars show the standard error of the mean, and stars indicate *p*-value < 0.05 using Student's unpaired *t*-test.

Figure EV3. Sex-specific differences of lipid abundance in rhesus monkeys. (A) Effect sizes (Males vs. Females) across lipid classes are plotted and colored by lipid category. Each point represents a lipid species, with point size proportional to mean log₂ abundance of the related feature/lipid species across all samples. Horizontal bars indicate mean effect size within each class. Negative values denote higher abundance in Females, while positive values denote higher abundance in Males. Lipid classes identified as significantly enriched or depleted by enrichment analysis are denoted with stars by name. (B) Averaged abundances of lipid classes significantly associated with sex are plotted as a function of age; title colors are based on the lipid category, dots show the mean abundance, error bars show the standard error of the mean, and stars indicate p -value < 0.05 using Student's unpaired t -test.

Appendix Figure S6. Age and age-associated differences of lipid abundance in rhesus monkeys. (A) Effect sizes based on the linear regression results for (Younger vs. Older monkeys) across lipid classes. (B) Effect sizes based on the linear regression results for age-diet interactions, and (C) Effect sizes based on the linear regression results for age-sex interactions. Effect sizes are plotted for each lipid class and colored by lipid category. Each point represents a lipid species that is colored and grouped by lipid category. Point size is proportional to the mean log₂ abundance of the related feature/lipid species across all samples. Horizontal bars indicate the mean of effect sizes within each class. Negative values denote higher abundance in Females, while positive values denote higher abundance in Males. Lipid classes identified as significantly enriched or depleted by enrichment analysis are denoted with stars.

6) *The relationship between sex dimorphism and aging is not adequately discussed. It is essential to provide relevant background information in the Introduction section.*

Response: Thank you for raising this important point that was also mentioned by reviewer 2. In response, we have expanded the Introduction to provide additional background on sex dimorphism in aging across mammalian species and its potential impact on lifespan regulation. See added text below to the **Introduction**:

“In addition to dietary and metabolic interventions, sex dimorphism is increasingly recognized as an important factor in aging biology. Across mammals, males and females frequently exhibit distinct lifespan trajectories, characterized by sex-specific differences in survival, disease susceptibility, and molecular aging signatures that impact the rate and nature of aging (Hägg & Jylhävä, 2021). These differences have been attributed to hormonal regulation, immune function, and metabolic pathways (Calabrò et al, 2023; Pomatto et al, 2018; Santos-Marcos et al, 2023). The importance of sex as a biological variable has gained traction in the biology of aging studies (Chen et al, 2022) and is expected to provide actionable insights of relevance to human aging.”

10th Nov 2025

Manuscript Number: MSB-2025-13100R

Title: Aging-linked Systemic Lipid Signature is Reprogrammed by Caloric Restriction in Rhesus Monkeys

Author: SALMA ABOUELHASSAN

Josef Clark

Di Kuang

Timothy Rhoads

Ricki Colman

Joshua Coon

Rozalyn Anderson

Katherine Overmyer

Dear Dr. Overmyer,

Thank you for sending us your revised manuscript. We have now heard back from the two reviewers who were asked to re-evaluate your study. As you will see, the reviewers are overall satisfied with the modifications made. Before we can formally accept your manuscript, we would ask you to address the following minor issues:

1. The remaining comments from Reviewer #3 regarding the inclusion of specific references in the Discussion.

On a more editorial level:

2. Please remove the figures from the manuscript file. Figure legends should be placed below the References section.

3. In your next submission, please complete the author information in the submission system, including institutional address, city, and phone number.

4. Please remove the synopsis image from the manuscript file.

5. Please reduce the number of keywords to five and place them below the Abstract.

6. Table EV1: Since this table is rather complex, the source file name, title, legend, and manuscript callout should all be updated to Dataset EV1 instead of Table EV1. The legend should be uploaded as a separate tab/sheet in the Excel file.

7. Appendix: Please use the nomenclature Appendix Tables S1-S3 consistently throughout the Appendix PDF, and ensure that table legends are included.

8. Data availability: please remove the reviewer credential statement and make sure the datasets will be made publicly available upon acceptance of the manuscript.

9. Please provide a "standfirst" to summarize the study in one or two sentences (approximately 250 characters, including space).

10. Source data:

- SD should be uploaded as one (zipped) file /figure, and named as "manuscriptID_SourceDataForFigure x"

- In the SD checklist, please untick the panels marked "others (R script)." Note that if the entire figure is supported by deposited large-scale data, there is no need to check individual panels.

11. Please address the following issues in figure legends:

- Please note that the exact p values are not provided in the legends of figures 4C, 5C, EV3 B

- Please indicate the statistical test used for data analysis in the legends of figure 4A

- Please note that information related to n is missing in the legends of figures 4A, C; 5C, 6A, C, E

12. "Materials and methods" should be renamed to "Methods".

13. The section titles and order should be corrected as follows: Title page - Abstract - Keywords - Introduction - Results - Discussion - Methods - Data Availability - Acknowledgements - Disclosure and Competing Interests Statement - References - Figure Legends - Table(s) - Expanded View Figure Legends.

Click on the link below to submit your revised paper.

Kind regards,
Jingyi

Jingyi Hou, PhD
Senior Editor
Molecular Systems Biology

*** PLEASE NOTE *** As part of the EMBO Press transparent editorial process initiative (see our Editorial at <https://dx.doi.org/10.1038/msb.2010.72> , Molecular Systems Biology will publish online a Review Process File to accompany accepted manuscripts. When preparing your letter of response, please be aware that in the event of acceptance, your cover letter/point-by-point document will be included as part of this File, which will be available to the scientific community. More information about this initiative is available in our Instructions to Authors. If you have any questions about this initiative, please contact the editorial office (msb@embo.org).

Reviewer #2:

I suggest the publication of the manuscript in its current form.

Reviewer #3:

I have reviewed the authors' responses and the revised manuscript. I am satisfied with the revisions made in response to the previous round of comments. The manuscript has been significantly improved.

I have no additional major concerns. However, the current discussion still lacks specific citations that directly connect the particular lipid classes or molecules found in this study to their established roles in aging processes from the literature. For instance, if certain phospholipids or sphingolipids were highlighted as key findings, the discussion should explicitly cite seminal or recent papers that have functionally demonstrated their involvement in aging pathways (e.g., in model organisms).

Therefore, I recommend that the authors further strengthen the discussion by incorporating more targeted references. This will solidly ground their extensive lipidomic dataset within the existing biological context of aging, as originally suggested.

Once this minor but important revision is made, I will be pleased to recommend the manuscript for acceptance.

Response to reviewers' comments

We thank all reviewers for taking the time to review our manuscript and for their thoughtful suggestions. Guided by those comments, we have revised the text and added new citations that frame the context for the work and the new insights derived to address minor comments raised by reviewer 3 and Editor Dr. Jingyi Hou. Specific comments are addressed below:

Remarks to the Author Reviewer #2: *I suggest the publication of the manuscript in its current form.*

Response: We sincerely thank the reviewer for their positive evaluation and support for publication of our manuscript in its current form.

Remarks to the Author Reviewer #3:

I have reviewed the authors' responses and the revised manuscript. I am satisfied with the revisions made in response to the previous round of comments. The manuscript has been significantly improved. I have no additional major concerns. However, the current discussion still lacks specific citations that directly connect the particular lipid classes or molecules found in this study to their established roles in aging processes from the literature. For instance, if certain phospholipids or sphingolipids were highlighted as key findings, the discussion should explicitly cite seminal or recent papers that have functionally demonstrated their involvement in aging pathways (e.g., in model organisms). Therefore, I recommend that the authors further strengthen the discussion by incorporating more targeted references. This will solidly ground their extensive lipidomic dataset within the existing biological context of aging, as originally suggested. Once this minor but important revision is made, I will be pleased to recommend the manuscript for acceptance.

Response: We appreciate the reviewer's suggestion and thank them for their positive assessment and for highlighting the need to further connect our findings to established literature on lipid regulation of aging. In the revised manuscript, we have strengthened the **Discussion** by explicitly citing key studies in model organisms that functionally link specific lipid classes to lifespan and aging pathways. See new text below:

“These patterns are consistent with lipidomic and genetic studies in model organisms that demonstrate that remodeling of sphingolipids and glycerolipids is tightly coupled with lifespan regulation. In *C. elegans*, the loss of *asm-3* (acid sphingomyelinase), which converts sphingomyelin to ceramide, extended lifespan and promoted stress resistance by downregulating the insulin/IGF-1 signaling (IIS) pathway and activating the longevity-associated transcription factor DAF-16/FOXO (Kim & Sun, 2012). Notably, long-lived *asm-3* mutant worms also have reduced TGs abundances, consistent with our observations in the CR monkeys with aging (Staab et al, 2023). Together these data suggest that disrupted sphingomyelin–ceramide balance impacts lifespan.”

25th Nov 2025

Manuscript number: MSB-2025-13100RR

Title: Aging-linked Systemic Lipid Signature is Reprogrammed by Caloric Restriction in Rhesus Monkeys

Dear Dr. Overmyer,

Thank you again for sending us your revised manuscript. We are now satisfied with the modifications made and I am pleased to inform you that your paper has been accepted for publication.

Sincerely,
Jingyi

Jingyi Hou, PhD
Senior Editor
Molecular Systems Biology
